# Pan-cancer association of DNA repair deficiencies with whole-genome mutational patterns

**Simon Grund Sørensen[1,2], Amruta Shrikhande[3], Gustav Alexander Poulsgaard[1,2], Mikkel Hovden Christensen[1,2], Johanna Bertl[1,2], Britt Elmedal Laursen[1,2,4], Eva R Hoffmann[3]\*, Jakob Skou Pedersen[1,2,5]\***

[1]Department of Molecular Medicine (MOMA), Aarhus University Hospital, Aarhus, Denmark; [2]Department of Clinical Medicine, Aarhus University, Aarhus, Denmark; [3]DNRF Center for Chromosome Stability, Department of Cellular and Molecular Medicine, University of Copenhagen, Copenhagen, Denmark; [4]Department of Biomedicine, Aarhus University, Aarhus, Denmark; [5]Bioinformatics Research Center (BiRC), Aarhus University, Aarhus, Denmark

**\*For correspondence:**
eva@sund.ku.dk (ERH);
jakob.skou@clin.au.dk (JSkouP)

**Competing interest:** The authors declare that no competing interests exist.

**Abstract** DNA repair deficiencies in cancers may result in characteristic mutational patterns, as exemplified by deficiency of *BRCA1/2* and efficacy prediction for PARP inhibitors. We trained and evaluated predictive models for loss-of-function (LOF) of 145 individual DNA damage response genes based on genome-wide mutational patterns, including structural variants, indels, and base-substitution signatures. We identified 24 genes whose deficiency could be predicted with good accuracy, including expected mutational patterns for *BRCA1/2*, *MSH3/6*, *TP53*, and *CDK12* LOF variants. *CDK12* is associated with tandem duplications, and we here demonstrate that this association can accurately predict gene deficiency in prostate cancers (area under the receiver operator characteristic curve = 0.97). Our novel associations include mono- or biallelic LOF variants of *ATRX*, *IDH1*, *HERC2*, *CDKN2A*, *PTEN*, and *SMARCA4*, and our systematic approach yielded a catalogue of predictive models, which may provide targets for further research and development of treatment, and potentially help guide therapy.

## Editor's evaluation

This is a well-motivated study looking at the association of DNA repair deficiencies with mutational patterns. This study is of interest to the cancer genomics community and highlights how the understanding of DNA repair processes can be used in the development of novel cancer therapy, and will also be of interest to researchers in the field of genomic medicine and cancer mutagenesis. It presents predictive models with potential clinical applications that can identify patients with specific gene dysfunction based on characteristic patterns of mutation. The key findings are well supported.

## Introduction

The DNA damage response (DDR) and repair pathways are central to the genetic integrity of cells, and deficiencies may cause mutational patterns genome-wide (*Lindahl, 1993*; *Nik-Zainal et al., 2012*; *Volkova et al., 2020*). Some DNA repair deficiencies are known to modulate the response to therapies: *BRCA1/2* deficiency renders cancers susceptible to treatment with PARP inhibitors (*Bryant et al., 2007*), mismatch repair (MMR)-deficient cancers are sensitive to checkpoint inhibitors (*Le et al., 2015*) but resistant to alkylating agents such as temozolamide (*von Bueren et al., 2012*), and

**eLife digest** Many different aspects of the environment – such as ultraviolet radiation, carcinogens in food and drink, and the ageing process itself – damage the DNA in human cells. Normally, cells can repair these sites by activating a mechanism known as the DNA damage response. However, the hundreds of genes that orchestrate this response are also themselves often lost or damaged, allowing the unrepaired sites to turn into permanent mutations that accumulate across the genome of the cancer cell.

By studying the DNA of cancer cells, it has been possible to identify characteristic patterns of mutations, called mutational signatures, that appear in different types of cancer. One specific pattern has been linked to the loss of either the *BRCA1* or *BRCA2* gene, both of which are part of the DNA damage response. However, it remained unclear how many other genes involved in the DNA damage response also lead to detectable mutational signatures when lost.

To investigate, Sørensen et al. computationally analysed data from over six thousand cancer patients. They looked for associations between over 700 DNA damage response genes and 80 different mutational signatures. As expected, the analysis revealed a strong connection between the loss of *BRCA1/BRCA2* and their known mutational signature. However, it also found 23 other associations between DNA damage response genes that had been lost or damaged and particular patterns of mutations in a variety of cancers. These findings suggest that mutational signatures could be used more widely to predict which DNA damage response genes are no longer functioning in the genome of cancer cells.

The mutational signature caused by the loss of *BRAC1/BRAC2* has been shown to make patients more responsive to a certain type of chemotherapy. Further experiments are needed to determine whether the connections identified by Sørensen et al. could also provide information on which treatment would benefit a cancer patient the most. In the future, this might help medical practitioners provide more personalized treatment.

*CDK12*-mutated cancers have a suggested sensitivity to CHK1 inhibitors (*Paculová et al., 2017*). Because of this, efforts have been made to annotate inactivating mutations in DDR genes (*Landrum et al., 2014*). However, the approach is limited by the lack of functional impact annotation of most variants, which are generally denoted as 'variants of unknown significance' (VUS). Moreover, loss of gene activity could also occur by other means, such as transcriptional silencing.

A complementary approach is to investigate whether DNA repair deficiencies can be identified by DNA mutational patterns, also referred to as 'mutational scars'. This approach has been pioneered for homologous recombination deficiency (HRD) caused by *BRCA1/2* deficiencies (-d), which can be successfully predicted by measuring the accumulation of small deletions with neighbouring microhomologous sequences (*Nik-Zainal et al., 2012*; *Davies et al., 2017*; *Nguyen et al., 2020*), such as done by the HRDetect algorithm by *Davies et al., 2017*. The association with microhomologous deletions is due to the use of microhomology-mediated endjoining to repair double-strand breaks in homologous recombination deficient tumours (*McVey and Lee, 2008*; *Nussenzweig and Nussenzweig, 2007*). Likewise, MMR deficiency causes an elevated rate of mono- and dinucleotide repeat indels genome-wide, a genetic phenotype denoted microsatellite instability (MSI; *Umar et al., 1994*; *Edelmann et al., 2000*). Mutations in other DNA repair genes have also been associated with mutational patterns, including the tumour suppressor gene *TP53*, which is associated with increased structural rearrangements and whole-genome duplications (*Lanni and Jacks, 1998*; *Gorgoulis et al., 2005*) and *CDK12* which is associated with a genome-wide phenotype of large tandem duplications (*Popova et al., 2016*; *Menghi et al., 2018*). The scope of this approach can now be evaluated systematically across DDR genes by exploiting available whole cancer genomes from thousands of patients (*ICGC/TCGA Pan-Cancer Analysis of Whole Genomes Consortium, 2020*; *Priestley et al., 2019*).

To achieve this, mutations observed genome-wide may be condensed into mutational summary statistics for predictive modelling, including statistics based on single base subsitutions (SBSs), indels, and different types of structural variants (SVs). The SBSs are statistically assigned to so-called SBS signatures that are catalogued and enumerated within the COSMIC database (*Tate et al., 2019*). Some of these are associated with specific DNA repair deficiencies as well as genotoxic exposures,

such as ultraviolet (UV) light and smoking. Each SBS signature captures the relative frequency of the different mutation types and their flanking nucleotides (*Alexandrov and Stratton, 2014*).

Here, we performed a systematic screen for DDR gene deficiencies that can be predicted through their association with genome-wide mutational patterns. We developed a generic approach to train predictive statistical models that identify associations with individual mutational summary statistics that capture the mutational patterns, including SBS signatures, indels, and large SVs. We applied it to 736 DDR gene deficiencies, considering both mono- and biallelic loss-of-function (LOF), identified across 32 cancer types, in a combined set of whole cancer genomes from 6065 patients (*ICGC/TCGA Pan-Cancer Analysis of Whole Genomes Consortium, 2020Priestley et al., 2019*). The underlying aim was to identify novel associations with potential biological relevance and to evaluate whether DDR deficiencies can be predicted with sufficiently high certainty to have a potential for clinical application.

Our analysis revealed 24 DDR genes where deficiencies are associated with specific mutational summary statistics in individual cancer types across 48 predictive models. These results recapitulated the expected associations between mutational patterns and deficiencies of *BRCA1/2*, *TP53*, *MSH3/6*, and *CDK12*. We supplemented this knowledge by providing a predictive model of *CDK12* deficiency that achieved high accuracy (area under the receiver operator characteristic [AUROC] = 0.97) in prostate cancer. Furthermore, we present unexpected predictive models of several DDR deficiencies; *ATRX* and *IDH1* deficiency in cancers of the central nervous systems (CNSs), *HERC2* and *CDKN2A* deficiency in skin, *PTEN* deficiency in cancers of the CNS and uterus, and *SMARCA4* deficiency in cancers of unknown primary.

## Results

### DDR gene deficiencies across 6065 whole cancer genomes

We compiled and analysed 2568 whole-genome sequences (WGS) from The Pan-Cancer Analysis of Whole Genomes (PCAWG) (*ICGC/TCGA Pan-Cancer Analysis of Whole Genomes Consortium, 2020*) and 3497 WGS from the Hartwig Medical Foundation (HMF) (*Priestley et al., 2019*). In total, we investigated 6065 whole cancer genomes of 32 cancer types (*Figure 1a*; *Supplementary file 1a*).

For each genome, we evaluated 736 known DDR genes for both germline and somatic LOF events (*Pearl et al., 2015*; *Knijnenburg et al., 2018*; *Olivieri et al., 2020*). We annotated both mono- and biallelic LOF events, where each event could be either a single-nucleotide variant, an indel, or a loss-of-heterozygosity (LOH) (*Figure 1b, c*; *Supplementary file 1b*). Pathogenicity of SBSs and indels was evaluated using a combination of CADD scores (>25; 0.3% most pathogenic variants) (*Rentzsch et al., 2019*) and ClinVar annotation, when available (Methods).

We inferred a total of 8408 biallelic DDR gene deficiencies, primarily through a combination of somatic or germline variants (SBSs and indels) with pathogenic potential ($n$ = 1702), or LOH events combined with a single pathogenic germline ($n$ = 3562) or somatic ($n$ = 3078) variant (SBS or indel; *Figure 1b*). On average we observed a single, biallelic DDR gene loss per patient, with some tumours showing extreme rates of somatic pathogenic mutations (*Figure 1c*; *Figure 1—figure supplement 1*).

As expected, *TP53* deficiency (*TP53*-d) was the most frequent LOF event (81 biallelic and 1746 monoallelic events; 29% of tumours affected; *Supplementary file 1b*; *Figure 1d*), while 70% of DDR genes had biallelic deficiency in less than 10 tumours across all cancer types (511/736; *Figure 1d*). Among monoallelic events, we identified 15,063 pathogenic germline (59%) and 10,336 somatic (41%) events.

### Whole-genome mutational patterns

We collected mutational summary statistics for each cancer genome, which were used as features for the downstream predictive models (*Figure 1e, f*; *Supplementary file 1c, d*). For SBSs, we evaluated exposure towards predefined sets of cohort-specific SBS signatures (*Alexandrov and Stratton, 2014*; *Degasperi et al., 2020*). Short indels and SVs were simply categorised and counted: Deletions were sub-categorised based on surrounding sequence repetitiveness and presence of microhomology. SVs were sub-categorised by type (tandem duplications, inversions, deletions, and translocations), five size ranges (not relevant for translocations), and cluster presence (Methods). Several SBS signatures as well as some types of indels have suggested aetiologies (collected in *Supplementary file 1e*).

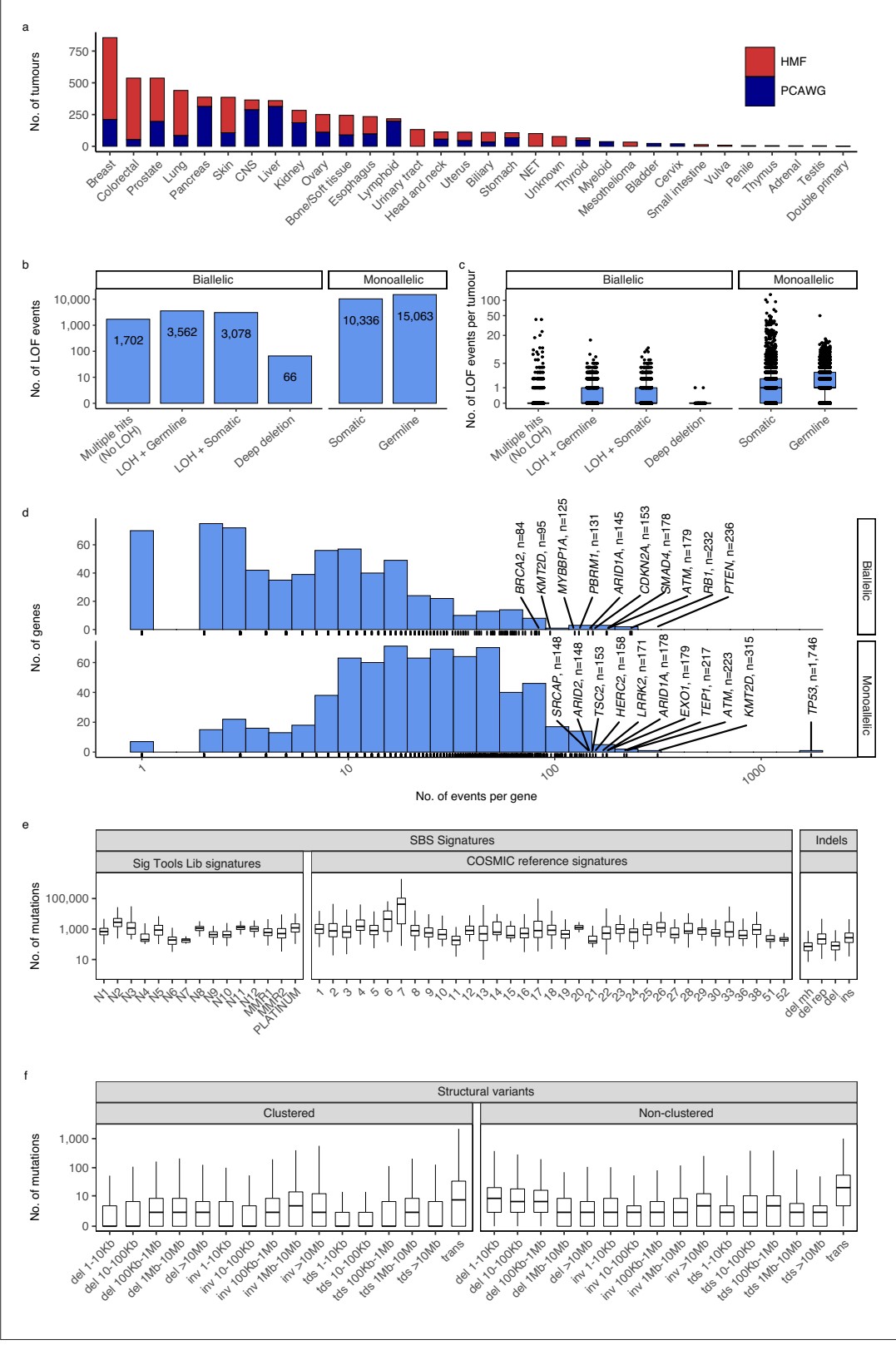

**Figure 1.** Cancer types, DNA damage response (DDR) gene deficiencies, and mutational patterns. (**a**) Cohort sizes for the 32 cancer types comprising the 6065 whole cancer genomes collected from the Hartwig Medical Foundation (HMF; *n* = 3497) and the PanCancer Analysis of Whole Genomes (PCAWG; *n* = 2568). (**b**) Mono- and biallelic loss-of-function (LOF) events were annotated across 736 DDR genes based on both pathogenic

*Figure 1 continued on next page*

*Figure 1 continued*

variants and copy number losses (loss of heterozygosity; LOH), overall and (**c**) per patient (**d**) with varying no. of LOF events per DDR gene (*x*-axis; logarithmic). (**e**) Whole-genome mutational patterns were represented as summary statistics and used as input features for the predictive models of DDR gene deficiency. Concretely, each patient was annotated with the number of single-base substitutions (SBSs) that are accounted to each SBS signature (*Alexandrov and Stratton, 2014*; *Degasperi et al., 2020*), number of indels divided by context (mh = microhomology; rep = repetitive), and (**f**) number of structural variants divided by clusterness, size, and type (del = deletion; inv = inversion; tds = tandem duplication; trans = translocation).

The online version of this article includes the following figure supplement(s) for figure 1:

**Figure supplement 1.** Loss-of-function events across tumours.

## Statistical modelling of DDR gene deficiencies

For the downstream statistical analysis, we restricted our focus to DDR genes in cancer types with more than five biallelic LOF events in either PCAWG or HMF (*n* = 194) or more than 10 monoallelic LOF events (*n* = 341) (*Supplementary file 1f*). Using *BRCA2*-d in the set of HMF breast cancer tumours (*n* = 645) as an example, we observed biallelic LOF events in 17 (2.6%; 14 germline, 3 somatic) tumours and monoallelic LOF events in 7 (1.1%; 4 germline, 3 somatic) tumours (*Supplementary file 1g*). We further observed VUS events in 53 tumours (8.2%; 42 germline, 11 somatic), which were excluded from the analysis. The remaining *BRCA2* wild-type (WT) tumours (*n* = 568; 88.1%) were used as a background set for training the predictive models (*Figure 2a*). The high fraction of germline pathogenic variants diminishes the probability of a reverse-causal relationship between the loss of *BRCA2* and the associated mutation patterns.

For each of the 535 groups of tumours we trained a least absolute shrinkage and selection operator (LASSO) regression model and evaluated the ability to discriminate between deficient and WT tumours (Methods). For *BRCA2*-d, we observed a strong association with the number of deletions at sites of microhomology (*Figure 2b*), with a median of 608 deletions per patient in *BRCA2*-d breast cancers versus 81 in *BRCA2* WT breast cancers, in agreement with prior findings (*Nik-Zainal et al., 2012*; *Davies et al., 2017*; *Nguyen et al., 2020*). The LASSO regression also included non-clustered inversions 10–100 kb and clustered tandem duplications 1–10 kb, although both show high variance among tumours for both deficient and WT (*Figure 2b*) and have considerably smaller coefficients, ultimately contributing little influence on overall predictive performance (*Figure 2c*).

Notably, some models include features with negative coefficients. The biological interpretation would be that tumours with a certain gene deficiency have fewer mutations attributed to a particular mutation pattern. Negative features were excluded in the development of the HRDetect algorithm (*Davies et al., 2017*), but we include them as we cannot rule out the possibility that a DDR deficiency protects from specific types of mutagenesis. Though not distinguishable in this study, we suggest that negative coefficient features may derive in three ways: First, they may stem from enhanced repair; second, they may stem from the decomposition of mutation counts into signatures; and third, the mutated tumours may represent a subclass of patients in terms of age, gender, or tumour subtype with specific mutational patterns.

## Evaluating model performance

For each model, we evaluated the predictive performance using the area-under-the-receiver-operating-curve (AUROC) score as well as the precision-recall area-under-the-curve (PR-AUC) score. The PR-AUC score is a more robust measure for unbalanced data sets (*Davis and Goadrich, 2006*); however, the expected value for non-informative (unskilled) models equals the fraction of true positives and thus varies between models. Therefore, we used the PR-AUC enrichment over the true-positive rate (PR-AUC-E) as our selection criteria for predictive models.

## Shortlisting models

For the downstream analysis, we included (shortlisted) models with PR-AUC-E that was substantial (>0.2; more than two standard deviations above the mean across all 535 models) and significant (Benjamin–Hochberg false discovery rate, FDR <0.05; Monte Carlo simulations) (*Figure 3*; *Supplementary file 1h*).

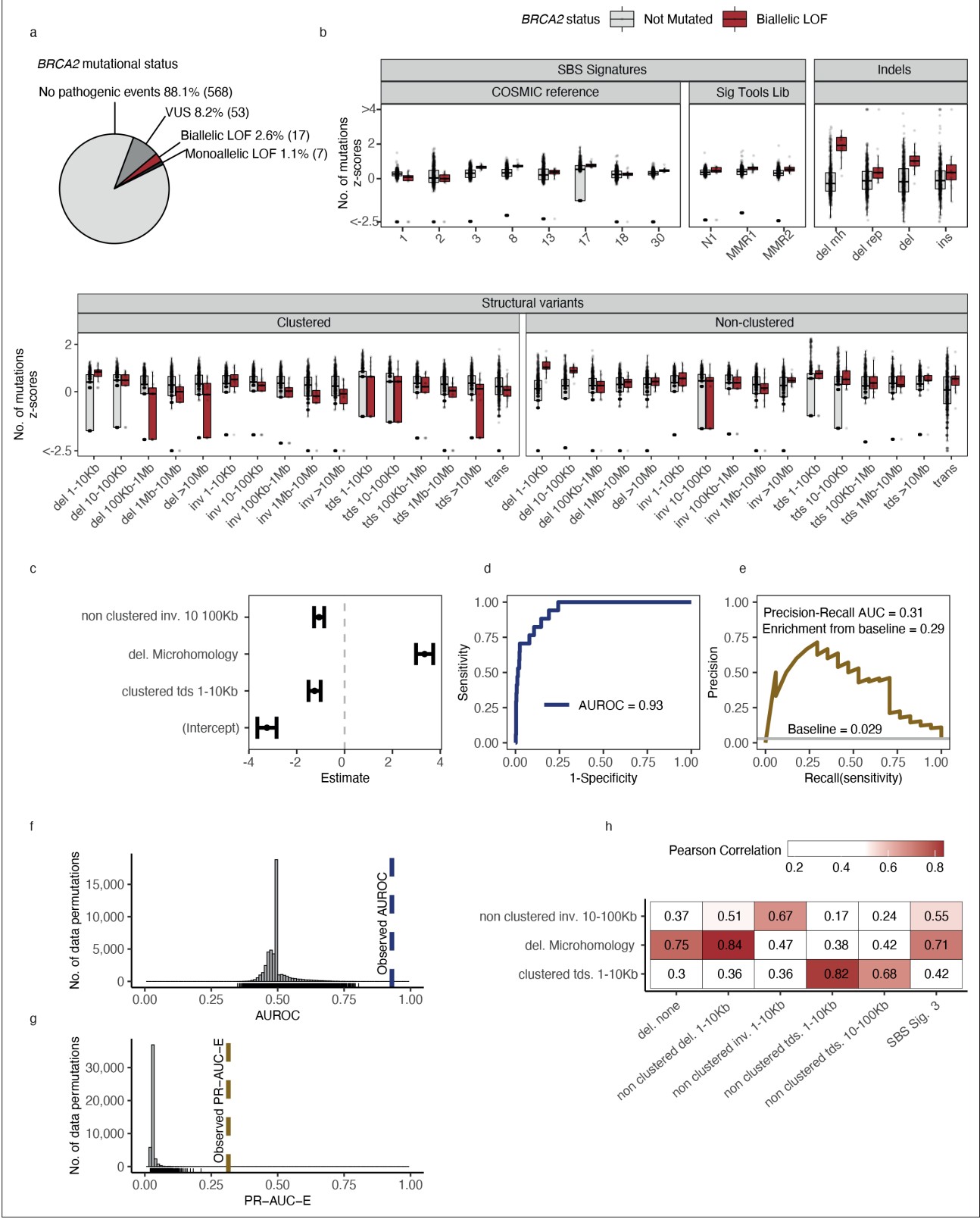

**Figure 2.** Predictive modelling of BRCA2 deficiencies in the Hartwig Medical Foundation (HMF) breast cancers. (**a**) Mutational status of *BRCA2* across 645 HMF breast cancer patients. (**b**) Mutational summary statistics for the HMF breast cancer patients divided by biallelic *BRCA2* loss-of-function (LOF; red) and *BRCA2* wild-type (WT; grey) (selected predictive features in bold). (**c**) Predictive features and their coefficients for model of biallelic *BRCA2* loss with predictive performance measured in (**d**) area under the receiver operator characteristic (AUROC) and (**e**) precision-recall area-under-the-

*Figure 2 continued on next page*

Figure 2 continued

curve (PR-AUC) (PR-AUC-E = PR-AUC − baseline = 0.29; Methods). (**f**) Distributions of AUROC and (**g**) PR-AUC-E values obtained from 30,000 random data permutations compared to observed values (punctuated lines). (**h**) Correlation between selected predictive features (horizontal) and other highly correlated (Pearson corr. >0.65) mutational features (vertical).

## Testing models in the opposite data set

Additionally, we calculated the PR-AUC-E of each model when applied to the same cohort in the opposite data set (*Figure 3—figure supplement 1*). Due to the difference in biology between the two sets, and low numbers of LOF mutated samples, we did not use this as model performance criteria but have included the PR-AUC-E values and p values from the tests (*Supplementary file 1h*). We identified significant predictive power, across both metastatic and primary cancers, for deficiency models of *BRCA1/2*, *TP53*, *CDK12*, *PTEN*, *ARID1A*, and *IDH1A*. Each case is described in the respective part of the results.

## BRCA example

In the example of *BRCA2*-d in breast cancers of the HMF data set, our model achieved an AUROC of 0.93 and a PR-AUC-E of 0.29 (*Figure 2d, e*; *Supplementary file 1h*). Out of 30,000 permuted LOF-sets, none had a similar or higher PR-AUC score and we considered the model significant with a p-value $<3 \times 10^{-5}$ (FDR adjusted *q*-value $<6 \times 10^{-4}$) (*Figure 2f, g*). The model achieved a PR-AUC-E of 0.37 when tested on the PCAWG data set, suggesting that the model may generalise across both metastatic and non-metastatic tumours. This was further supported by the independent discovery of a similar model in the PCAWG data set, which had an almost similar predictive power in the HMF data (PR-AUC-E = 0.19; *Supplementary file 1h*). Notably, the *BRCA2*-d model did not include non-clustered deletions <100 kb, SBS signature 3, and SBS signature 8, all features which have been associated with BRCAness (*Davies et al., 2017*). However, SBS signature 3 and non-clustered deletions 1–10 kb are included in the model when the deletions at sites of microhomology are omitted from the input data set, suggesting that they are excluded during feature selection due to high positive correlation with the number of deletions at sites of microhomology among HMF breast cancers (Pearson corr. >0.7; *Figure 2h*; *Supplementary file 1i*).

Our selection criteria resulted in 48 shortlisted predictive models across 24 DDR genes (*Figure 3a*; *Supplementary file 1h*). As exemplified for *BRCA2*, each model is specified by a set of predictive features representing mutational patterns associated with DDR gene LOF. We divided the models into four groups based on aetiology and origin: models of *BRCA1/2*-d (eight models of *BRCA2*-d and a single model of *BRCA1*-d; *Figure 3b*); models of monoallelic *TP53*-d (11 models; *Figure 3c*); models of various monoallelic gene deficiencies derived from colorectal cancer patients (eight models; *Figure 3d*); and models including other DDR genes and cancer types, including previously undescribed associations (20 models; *Figure 3e*).

## Survival analysis

For each of the shortlisted models, we evaluated the difference in overall survival between samples carrying LOF mutations and those that did not. We observed nominally significant differences (p < 0.05; univariate Cox regression analysis) in survival for *BRCA2* and *TP53* in multiple cancer types as well as for *UVRAG* in colorectal cancer (*Figure 3—figure supplements 2 and 3*; *Supplementary file 1j*). The association of *TP53* monoallelic LOF with decreased survival is in line with expectations (*Malcikova et al., 2009*). Interestingly, several models of *BRCA1/2* LOF mutations associated with improved survival, including *BRCA1* LOF mutations in metastatic ovary cancers (hazard-ratio <0.42; p < 0.093) and *BRCA2* LOF mutations in non-metastatic ovary cancers (hazard-ratio <0.24; p < 0.017). In contrast, *BRCA2* LOF mutations in primary breast cancers were associated with decreased survival (hazard-ratio >9.30; p < 0.004) (*Figure 3—figure supplements 2 and 3*). This may potentially reflect differences in both molecular diagnostic practices and treatment regiments across these cancer types. For instance, platin-based treatment irrespective of *BRCA1/2* status has been standard for groups of the ovarian and pancreatic cancer patients, while traditionally not for the breast cancer patients (*Gennari et al., 2021*; *Colombo et al., 2019*). The sensitising effect of *BRCA1/2* deficiency might thus explain the associated survival differences among cancer types (*Kennedy et al., 2004*). For most

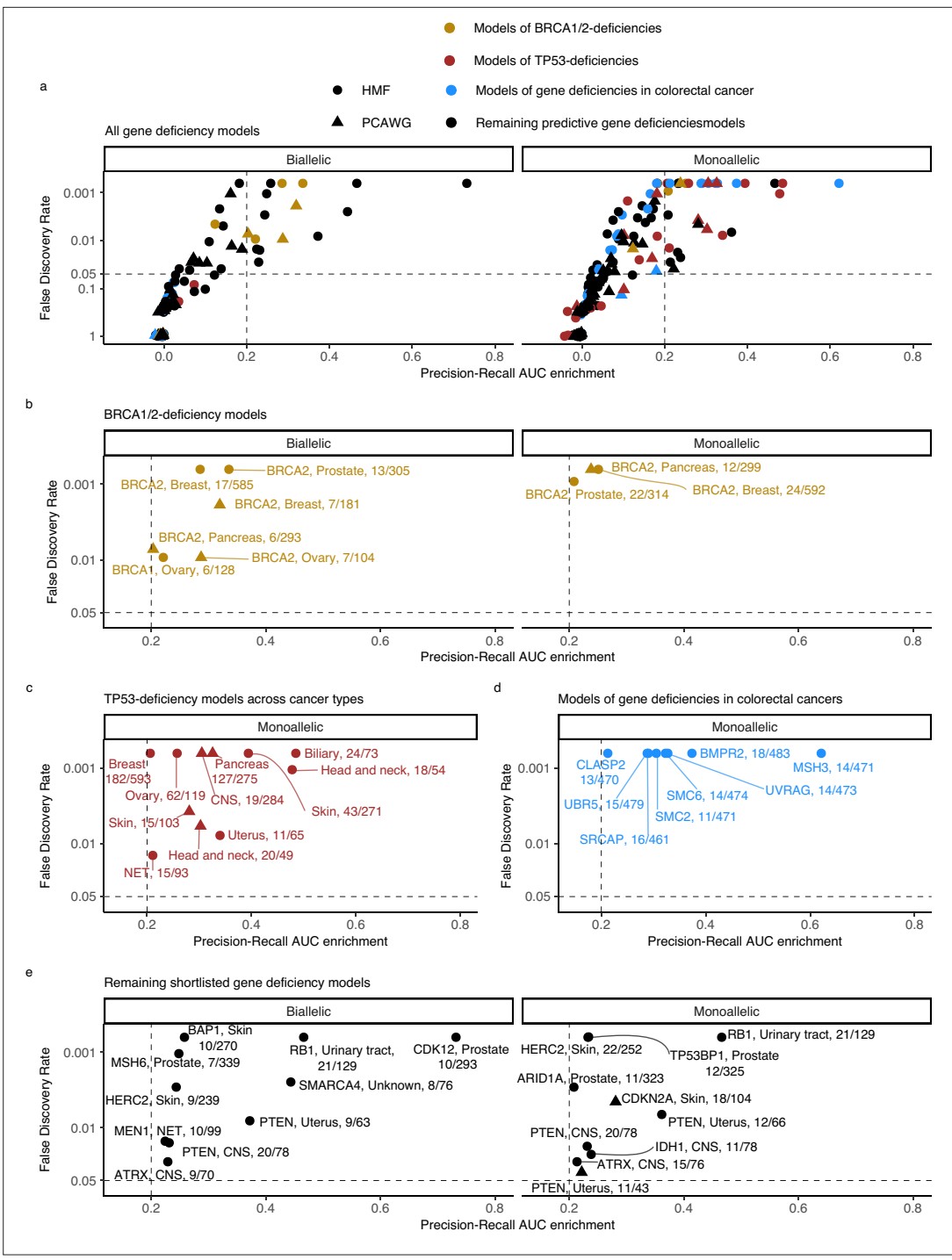

**Figure 3.** Predictive models of DNA damage response (DDR) gene deficiencies. (**a**) The precision-recall AUC enrichment PR-AUC-E; *x*-axis and significance (false discovery rate [FDR]; logarithmic *y*-axis) of the 535 predictive models (one model per gene with more than 5 biallelic or more than 10 tumours either mono- or biallelic mutated in either Hartwig Medical Foundation (HMF) or The Pan-Cancer Analysis of Whole Genomes (PCAWG) in any one cancer type; Methods). Significance (*q*-value representing FDR) evaluated by counting equally or more-extreme PR-AUC-E values across >10,000 permuted data sets and applying Benjamini–Hochberg FDR control. Models with FDR below 0.05 and PR-AUC-E above 0.2 are shortlisted (Methods). (**b**) Shortlisted predictive models of deficiency of *BRCA1* or *BRCA2*; (**c**) *TP53* monoallelic predictive models; (**d**) monoallelic gene deficiency models across colorectal cancer patients; and (**e**) remaining gene deficiency models not contained in the other sub-groups.

*Figure 3 continued on next page*

*Figure 3 continued*

Numbers indicate the number of mutated out of the total number of tumours included in the development of each model.

The online version of this article includes the following figure supplement(s) for figure 3:

**Figure supplement 1.** Evaluating model performance in the opposite data set.

**Figure supplement 2.** Survival analysis of patients with or without loss-of-function (LOF) events in shortlisted DNA damage response (DDR) genes.

**Figure supplement 3.** Kaplan–Meier survival plots for patients from cancer-type cohorts used to train the 48 shortlisted models.

---

models the differences in survival were insignificant, though this may be related to the generally small set of LOF mutated samples.

## Recapitulation and predictive modelling of expected associations with *BRCA1/2* deficiency

Five models predicted biallelic loss of *BRCA2* in cancers of the ovary, prostate, pancreas, and breast. In addition, three models predicted *BRCA2* monoallelic loss in cancers of the pancreas, breast, and prostate. Finally, we derived a single model of biallelic *BRCA1* loss in ovarian cancer (*Figures 3c and 4a*). All models significantly outperformed their Monte Carlo simulations (*q* < 0.05; Benjamini–Hochberg FDR control) and had PR-AUC-E above 0.2 (*Figure 4b, c*).

All *BRCA2*-d models were predominantly predicted by deletions at sites of microhomology, consistent with the role of *BRCA2* in homologous recombination and suppression of microhomology-mediated endjoining (*Ceccaldi et al., 2015*). Both clustered and non-clustered tandem duplications in the range of 1–100 kb were included as features for various models, though with much smaller predictive power. This agrees with what was identified for *BRCA2*-deficient tumours in prior studies (*Davies et al., 2017*; *Nguyen et al., 2020*). The biallelic breast cancer model based on PCAWG further included SBS signature 3 as a predictive feature (*Figure 4d*). Contrasting to the models of *BRCA2*-d, *BRCA1*-d in ovarian cancer was exclusively associated with clustered and non-clustered tandem duplications (1–10 kb; *Figure 4d*). This aligns with prior studies (*Davies et al., 2017*; *Nguyen et al., 2020*), which also found *BRCA1*-d to be closely associated with a tandem-duplicator phenotype. In general, *BRCA1* and *BRCA2* were subject to predominantly germline pathogenic events, and not a single deletion at a site of microhomology, suggesting the expected forward causality (*Supplementary file 1g*). As for the loss of *BRCA2*, the model for loss of *BRCA1* loss in ovary had sufficient predictive power (PR-AUC-E = 0.3) in the other data set, suggesting that the model works independently of the metastatic capacity of the tumour (*Supplementary file 1h*).

## *TP53* deficiencies associate with increased numbers of SVs

We detected 11 predictive models (four based on PCAWG and seven on HMF) of monoallelic *TP53*-d across cancers of the breast, skin, ovary, uterus, neuro-endocrine tissues, biliary gland, head and neck, pancreas, and the CNS (*Figure 4e*). These predictive models performed with PR-AUC-E values ranging from 0.21 to 0.48 in breast and biliary gland cancers, respectively. Similarly, AUROC values ranged from 0.48 to 0.88, again in breast and biliary gland cancers (*Figure 4f, g*). In line with existing literature (*Hanel and Moll, 2012*), *TP53*-d is associated with a significantly increased number of SVs across the genome (Wilcoxon rank-sum test; *Figure 4h, i*), except in skin cancers. The models of *TP53* loss in head and neck, skin, breast, and the biliary gland performed well (PR-AUC-E above 0.2) in the other data set, suggesting that the predictive performance generalises independent of metastatic tumour state (*Figure 3—figure supplement 1*; *Supplementary file 1h*).

## Colorectal cancer models derived from hypermutated MMR-deficient tumours

In the HMF colorectal cancers, we discovered eight predictive gene deficiency models (*MSH3*, *SMC2*, *SMC6*, *BMPR2*, *CLASP2*, *SRCAP*, *UBR5*, and *UVRAG*) of monoallelic LOF with PR-AUC-E ranging from 0.21 (*CLASP2*) to 0.62 (*MSH3*) (AUROC ranging from 0.68 for *SRCAP* deficiency to 0.94 for *MSH3* deficiency; *Figure 4j–l*).

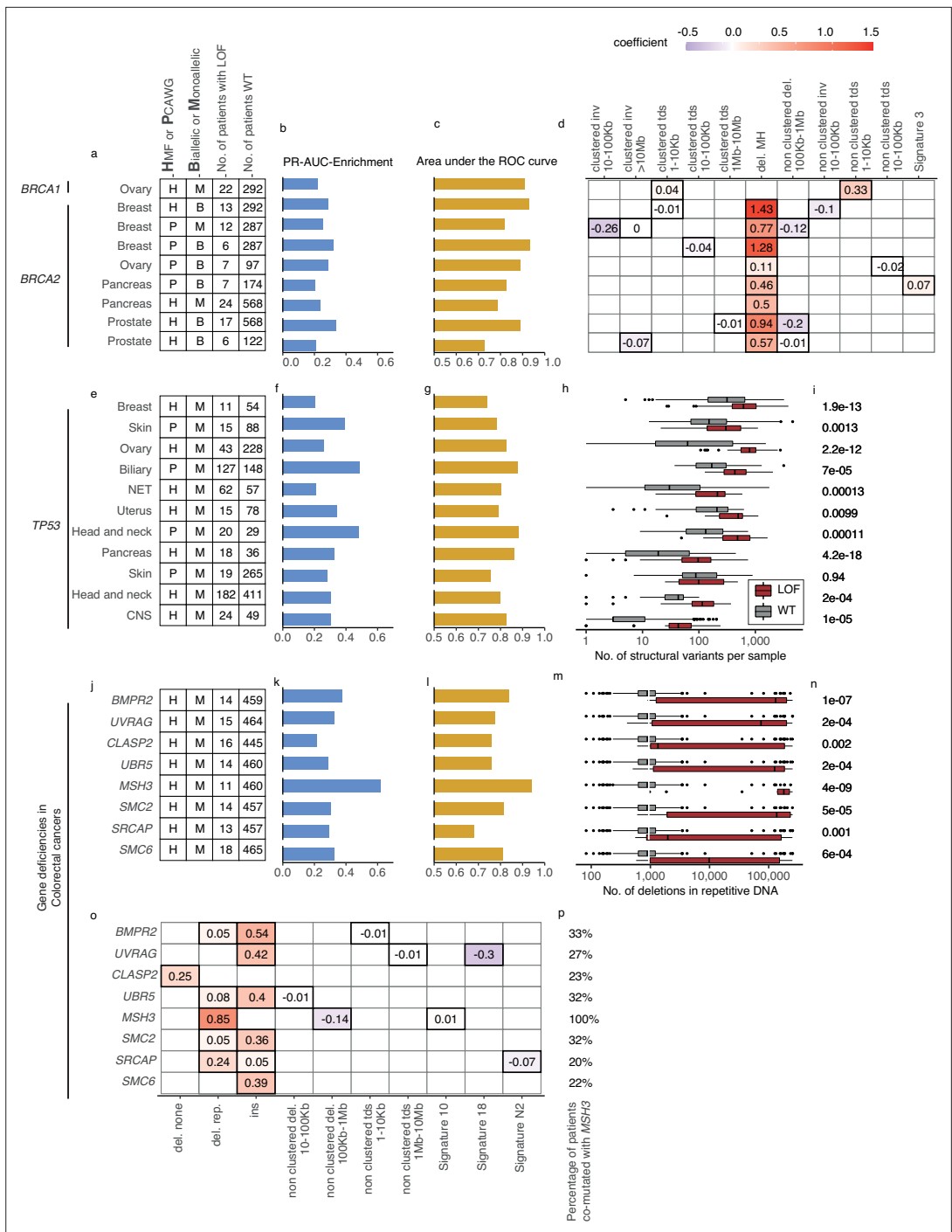

**Figure 4.** Predictive models with anticipated aetiology or origin. (**a**) Overview of predictive models for *BRCA1*-d and *BRCA2*-d, showing data source, type of model, and loss-of-function (LOF)-set statistics. (**b**) PR-AUC-E, (**c**) area under the receiver operator characteristic (AUROC), and (**d**) the predictive features and their coefficient for individual models. (**e**–**g**) Overview of predictive models of *TP53*-d (as in a–c). (**h**) For each cohort, the number of structural variants (x-axis; logarithmic) for *TP53* LOF tumours (red) versus *TP53* wild-type tumours (grey) and (**i**) the significance of their difference (two-sided Wilcoxon rank-sum test). (**j**–**l**) Predictive models of gene deficiencies in colorectal cancers (as in a–c). (**m**) Number of deletions in repetitive DNA (as in h) and (**n**) its significance (as in i). (**o**) The predictive features of each model (as in d) and (**p**) the percentage of tumours that are co-mutated with *MSH3*.

We suggest that the high number of models of monoallelic deficiencies may arise from spurious LOF events in DDR genes in a subset of colorectal cancers that are hypermutated. In line with this, the hypermutated samples ($n = 18$; >100,000 mutations) harbour 22% (5.9-fold enrichment) of all the DDR LOF events across the HMF colorectal cancer samples ($n = 475$).

Some colorectal cancers are signified by MMR deficiencies, such as LOF of *MSH3* or *MSH6*, creating a high number of deletions in repetitive DNA (*Umar et al., 1994*; *Edelmann et al., 2000*). Indeed, we found that this pattern was most profound among the *MSH3*-mutated cancers (*Figure 4m, n*). Furthermore, we found co-mutation with *MSH3* across the tumours underlying each model, ranging from 20% (*SRCAP*) to 33% (*BMPR2*) of the mutated tumours (*Figure 4p*). This suggests that the models (except for the model of *MSH3*-d) might be the consequence of the hypermutator pheno-type. In other words, the causality may be reversed in these cases, and the mutational process driven by *MSH3*-d may have caused the majority of their LOF events. This notion is supported by investigating the features of the models. All eight models are characterised by a single, primary predictive feature: Insertions (*SMC2*, *SMC6*, *BMPR2*, *UBR5*, and *UVRAG*), deletions in repetitive DNA (*MSH3* and *SRCAP*), or deletions not flanked by repetitive or microhomologous DNA (*CLASP2*) (*Figure 4o*). Each of these features has a high positive correlation (Pearson corr. >0.93) with the number of dele-tions in repetitive DNA. This correlation suggests that all eight models relate to a genome-instability phenotype, which may be driven by the *MSH3* co-mutated tumours or, potentially, a concurrent deficiency of other genes within the MMR system (*Supplementary file 1i*). Notably, the deficiency models of *UBR5*, *BMPR2*, *CLASP2*, and *SMC6* all had PR-AUC-E above 0.2 in the other data set, suggesting that these genes are associated with the MMR phenotype regardless of metastatic state (*Supplementary file 1h*).

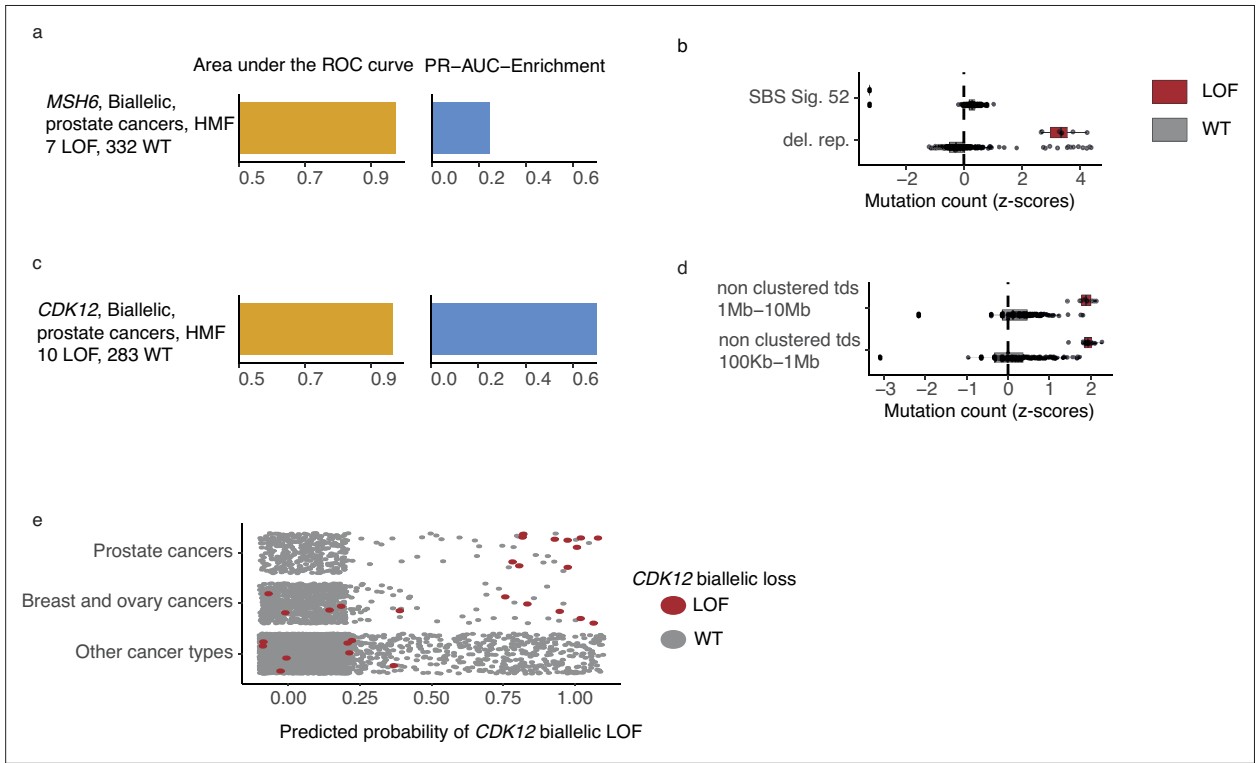

**Figure 5.** *CDK12*-mutated prostate tumours are predicted by tandem duplications. (**a**) Biallelic predictive model *MSH6*-d in Hartwig Medical Foundation (HMF) prostate tumours and its PR-AUC-E and area under the receiver operator characteristic (AUROC). (**b**) Boxplots of mutation counts between tumours that are *MSH6*-d loss-of-function (LOF; red) and *MSH6* wild-type (grey), (mutation counts are normalised and log-transformed; Methods). (**c**) Biallelic predictive model for *CDK12*-d with performance measures (as in a). (**b**) Boxplots of mutation counts for *CDK12*-d and wild-type tumours (as in b). (**e**) *CDK12*-d predictive performance for different cancer types. Predicted probability of *CDK12*-d (x-axis) for tumours with *CDK12* LOF (red) and *CDK12* wild-type (grey) are shown for prostate cancer, breast and ovary cancer, and all other cancer types (y-axis).

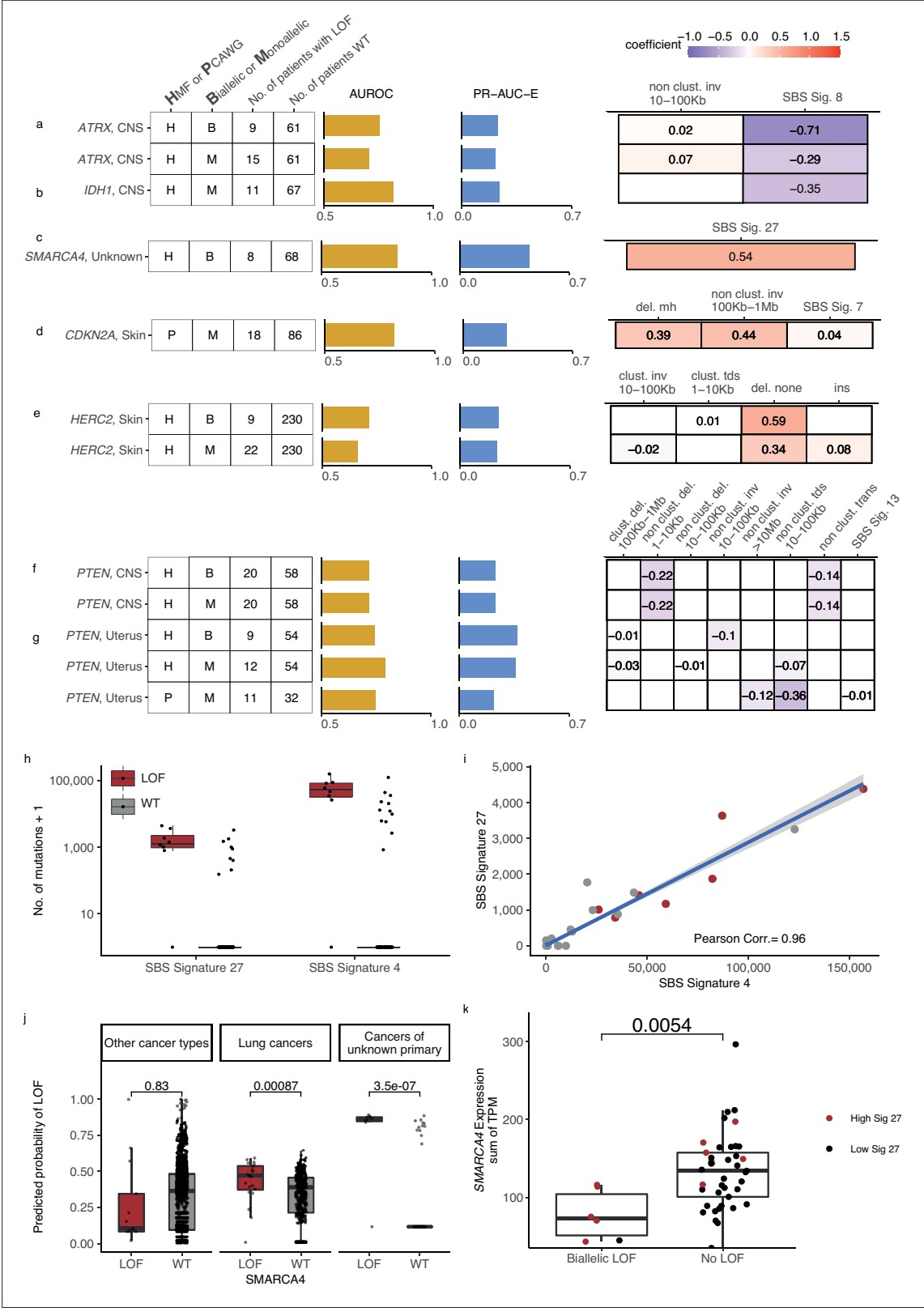

**Figure 6.** Novel predictive models of DNA damage response (DDR) gene deficiencies. (**a**) Predictive model of *ATRX*-d and its PR-AUC-E, area under the receiver operator characteristic (AUROC), and selected features and their coefficients. Same information for predictive models of (**b**) *IDH1*-d, (**c**) *SMARCA4*-d, (**d**) *CDKN2A*-d, (**e**) *HERC2*-d, and (**f**) *PTEN*-d in central nervous system (CNS) cancers and (**g**) uterine cancers. (**h**) Number of single-base substitution (SBS) sig. 27 and SBS sig. 4 (*y*-axis; logarithmic) among tumours of unknown primary with *SMARCA4* biallelic loss-of-function (LOF) (red)

*Figure 6 continued on next page*

*Figure 6 continued*

or wild-type (grey). (**i**) Pearson correlation between the per-tumour number (tumours of unknown primary; Hartwig Medical Foundation [HMF]) of SBS signature 27 (*y*-axis) and SBS signature 4 (*x*-axis; logarithmic) mutations, with an overlaid linear model (blue) and its 95% confidence interval (grey). (**j**) Using a model trained to predict *SMARCA4* biallelic LOF in HMF cancers of unknown primary, we evaluate the predictive power across individual cohorts (one-tailed Wilcoxon rank-sum test), displaying significant cohorts separately (colours as in **h**). (k) Expression of *SMARCA4,* meassured as the sum of all annotated transcripts per milion (TPM; y-axis), for tumours with biallelic LOF and no LOF (x-axis). Colors indicate the rate of SBS sig. 27 in each tumour, (red >0; black = 0). The difference in expression was evaluated using a non-paired Wilcoxon rank-sum test.

The online version of this article includes the following figure supplement(s) for figure 6:

**Figure supplement 1.** Additional shortlisted predictive models of DNA damage response (DDR) gene deficiencies.

**Figure supplement 2.** Monoallelic *CDKN2A* deficiency (-d) in The Pan-Cancer Analysis of Whole Genomes (PCAWG) and Hartwig Medical Foundation (HMF) skin cancers, mono- and biallelic *HERC2* deficiency in HMF skin cancers.

## Biallelic LOF of *MSH6* associated with increased number of deletions in repetitive DNA in prostate cancer

*MSH6*, a gene implicated in MMR and microsatellite stability (*Edelmann et al., 2000*), was mutated in both alleles in 7 out of 342 HMF prostate cancer patients. We observed pathogenic indels in *MSH6* in all seven tumours, but only one of these in mono- or dinucleotide repeat DNA. *MSH6* deficiency could be predicted with high accuracy (PR-AUC-E = 0.25; AUROC = 0.98) by an enrichment of deletions in repetitive DNA (*Figure 5a, b*; *Supplementary file 1h*). This is consistent with existing findings of MMR deficiency and its presence in metastatic prostate cancers (*Graham et al., 2020*).

## High predictive power for *CDK12* deficiency

*CDK12* encodes a kinase that regulates transcriptional and post-transcriptional processes of the DDR (*Blazek et al., 2011*; *Marqués et al., 2000*; *Li et al., 2016*). We found that *CDK12*-d prostate cancers had an increased number of mid- and large-sized tandem duplications 100 kb to 10 Mb, compared to *CDK12*-WT (*Figure 5c, d*). Several studies have observed similar tandem duplication phenotypes in ovarian cancers (*Popova et al., 2016*; *Menghi et al., 2018*; *Li et al., 2020*) and castration resistant prostate cancers (*Wu et al., 2018*; *Rescigno et al., 2021*). In agreement, nine of the 10 patients in our data set were treated with drugs associated with castration resistance (4 Enzalutamide, 3 Abiraterone, 1 Cabazitaxel, 1 Pembrolizumab) (*Sumanasuriya and De Bono, 2018*).

Whereas the tandem duplication phenotype has been previously associated with loss of *CDK12*, in this study we present the first predictive algorithm utilising and quantifying the high predictive value of these patterns (PR-AUC-E = 0.73 and AUROC = 0.97). Indeed, the loss of *CDK12* has been demonstrated to sensitise cancer cells to CHK1— (*Paculová et al., 2017*) and PARP inhibitors (*Bajrami et al., 2014*; *Joshi et al., 2014*).

We went on to test the *CDK12*-d model across other cancer types (*Figure 5c*). As expected, we observed predictive power in cancers of the ovary and breast, though at a lower level (PR-AUC-E = 0.19 and AUROC = 0.72). No predictive power was observed for the remaining cancer types. We only observed a single tumour with biallelic LOF of *CDK12* in PCAWG, but the predictive model was able to correctly identify this tumour, and reached a PR-AUC-E of 0.99 in the PCAWG data set.

## Novel predictive gene deficiency models

The shortlisted predictive LOF models further include biallelic LOF of *ATRX*, *PTEN*, *HERC2*, *MEN1*, *SMARCA4*, *BAP1*, and *RB1*, as well as monoallelic LOF of *ATRX*, *IDH1*, *PTEN*, *CDKN2A*, *ARID1A*, *TP53BP1*, *HERC2*, and *RB1* (*Figure 3e* and *Figure 6*; *Figure 6—figure supplement 1*). The number of mutated tumours underlying each model varied from 8 (biallelic *SMARCA*-d in cancers of unknown primary) to 22 (monoallelic *HERC2*-d in skin).

## Predictive models of *ATRX*-d and *IDH1*-d in CNS cancers

We found that *ATRX*-d (monoallelic: PR-AUC-E = 0.21, AUROC = 0.71; biallelic: PR-AUC-E = 0.23, AUROC = 0.76; *Figure 6a*) and *IDH1*-d (monoallelic: PR-AUC-E = 0.24, AUROC = 0.82; *Figure 6b*) in CNS cancers were both predicted by a decreased number of SBS signature 8 mutations. In addition, *ATRX*-d was further predicted by non-clustered inv. 10–100 kb, although with a small coefficient and hence contributing limited discriminatory power. We discovered that 7 of 11 *IDH1*-mutated tumours

were also *ATRX*-mutated (28-fold enrichment compared to *IDH1* WT skin cancers; p = 1.2 × 10⁻⁸, Fisher's exact test) and that both *ATRX* and *IDH1* were predominantly hit by somatic mutations (14 of 15 for *ATRX*, 7 of 11 for *IDH1*).

Co-mutation between *IDH1* and *ATRX* is well described in gliomas (*Mukherjee et al., 2018*). Interestingly, LOF of either gene is associated with lack of SBS signature 8, a signature associated with BRCAness (*Davies et al., 2017*; *Alexandrov et al., 2020*) and late-replication errors (*Singh et al., 2020*). This suggests that CNS cancers with *IDH1/ATRX* deficiency are not subject to the same DNA lesions or repair processes as other CNS cancers and they may potentially belong to a separate patient subclass, though we could not identify evidence of this.

## Predictive model for *SMARCA4*-d in cancer of unknown primary

We discovered eight tumours (HMF) out of 77 with cancers of unknown primary with *SMARCA4*-d (biallelic) that could be predicted with relatively high accuracy (PR-AUC-E = 0.44; AUROC = 0.85; *Figure 6c*). These tumours showed an enrichment of SBS signature 27 [a signature first detected in myeloid cancers (*Alexandrov et al., 2015*)], which has been suggested to be a sequencing artefact though it also displays strong strand bias (*Tate et al., 2019*; *Supplementary file 1e*). Among cancers of unknown primary, SBS signature 27 correlates strongly with SBS signature 4 (Pearson corr. = 0.96), despite the signatures different composition (cosine similarity of signatures = 0.17) (*Figure 6h, i*; *Supplementary file 1i*). This suggests that SBS signature 4 may also be associated with *SMARCA4*-d and we indeed found that its predictive performance (PR-AUC-E of 0.43; AUROC = 0.83) was almost equivalent to SBS signature 27. Signature 4 is associated with smoking across several cancer types (*Alexandrov et al., 2013*; *Nik-Zainal et al., 2015*); interestingly, *SMARCA4*-d is seen in aggressive thoracic sarcomas (*Sauter et al., 2017*) and strongly enriched among patients with a history of smoking (*Rekhtman et al., 2020*). Its gene product, BRG1, has been suggested as a lung cancer transcriptional regulator of genes that induce tumour proliferation (*Dagogo-Jack et al., 2020*) and metastasis (*Concepcion et al., 2022*). We evaluated the ability of SBS signature 4 to predict *SMARCA4*-d in other cancer types and found a significant predictive association in lung cancer, though much lower than for cancers of unknown primary (*Figure 6j*). We also found a significant ability to predict *SMARCA4*-d by the number of SBS signature 27 mutations in cancers of the neuro-endocrine tissues (Wilcoxon test, one-tailed p = 0.002) and head and neck (p = 4.8 × 10⁻⁷), but we could not evaluate this in lung cancers, as the signature is not among its set of cohort-specific signatures.

Twelve cancers had a high posterior probability of *SMARCA4*-d (*Figure 6j*) despite being *SMARCA4* WT and lacking pathogenic events. No other single DDR gene was mutated among all 12 tumours, with *TP53* having the most LOF events (6 of 12 cancers).

We evaluated the expression of *SMARCA4* among the cancers of unknown primary and identified a significantly lower expression of *SMARCA4* in tumours with biallelic LOF mutations compared to WT (p = 0.54 × 10⁻²; *Figure 6k*). For the 12 tumours with high signature 27 exposure but no biallelic LOF mutations, we did not observe a similar decrease in expression (*Figure 6k*), suggesting that *SMARCA4* was not epigenetically silenced or otherwise transcriptionally inactivated in these cases.

Given that cancers of unknown primary have disparate origins, this raises the possibility that the patients with high posterior probability of *SMARCA4*-d may have metastasised from a cancer type or subtype with both high levels of SBS signatures 4 and 27, as well as high incidence of *SMARCA4*-d. However, for the lung cancer samples, we did not observe any significant association of *SMARCA4-d* with subtype (p = 0.72 for HMF; p = 0.65 for PCAWG; Fisher's exact test). Further studies are thus needed to clarify if the observed associations can be explained through such an ascertainment bias rather than causatively by *SMARCA4*-d.

## Predictive model for monoallelic *CDKN2A*-d in skin cancer

Both germline and somatic variants in *CDKN2A* are known to predispose for melanoma (*Liu et al., 1999*). In the PCAWG skin cancer cohort, we found that a monoallelic predictive model of *CDKN2A*-d achieved relatively high accuracy (PR-AUC-E = 0.28; AUROC = 0.82). Its predictive features are enrichment of deletions at sites of microhomology, non-clustered inv. (100 kb to 1 Mb), and SBS signature 7, in order of predictive importance (*Figure 6d*). Apart from the inversions, these features are also significantly associated with biallelic *CDKN2A*-d in the HMF skin cancer cohort (*Figure 6—figure supplement 2a, b*) and are included in the corresponding predictive model, though it had lower predictive

performance (PR-AUC-E = 0.008; AUROC = 0.542) and was not shortlisted (*Supplementary file 1f*). In both HMF and PCAWG, most of the observed LOF events are somatic, with the majority being biallelic (16 of 18 in PCAWG; 32 of 33 in HMF; *Supplementary file 1g*). The presence of deletions at sites of microhomology suggests a possible reduction in error-free double-stranded break repair in combination with an increased accumulation of SBS signature 7.

## Predictive model for *HERC2-d* in metastatic skin cancer

*HERC2*-d has been associated with susceptibility to developing melanoma (*Amos et al., 2011*). We found *HERC2*-d predictable in HMF skin cancer patients (biallelic: PR-AUC-E = 0.24, AUROC = 0.73; monoallelic: PR-AUC-E = 0.23, AUROC = 0.66; *Figure 6e*), primarily based on enrichment of deletions in non-microhomologous and non-repetitive regions.

HERC2 encodes a protein ligase that modulates the activity of P53 (*Cubillos-Rojas et al., 2014*). We observed that seven of the nine tumours with biallelic *HERC2*-d also had a monoallelic pathogenic event in *TP53* (eightfold enrichment; p = $4.6 \times 10^{-6}$, Fisher's exact test). Tumours that are co-mutated in *TP53* (monoallelic) and *HERC2* (mono- or biallelic) showed a significantly higher number of deletions compared to tumours with LOF in either gene alone (Wilcoxon test, one-tailed p < 0.031) and cancers that are WT for both genes (p < $7.2 \times 10^{-7}$) (*Figure 6—figure supplement 2c*).

## *PTEN* deficiency associates with fewer SVs in CNS and uterine cancers

*PTEN* is a tumour suppressor gene found in various cancer types (*Li et al., 1997*; *Liaw et al., 1997*) and its deficiency was found to be associated with mutational patterns in uterine and CNS cancers (*Supplementary file 1h*). In CNS, we acquired two identical models from the HMF data set, as we observed no monoallelic events without a second hit (mono- and biallelic PR-AUC-E = 0.37, AUROC = 0.74; *Figure 6f*). In uterine cancer, we acquired significant predictive models from both HMF and PCAWG (HMF, biallelic: PR-AUC-E = 0.37, AUROC = 0.74; HMF, monoallelic: PR-AUC-E = 0.36, AUROC = 0.79; PCAWG, monoallelic: PR-AUC-E = 0.22, AUROC = 0.74; *Figure 6g*). In addition, the models of *PTEN* loss in uterus had predictive power in the other data sets, suggesting signal robustness and independence of metastatic state (the PCAWG-derived model had PR-AUC-E of 0.55 in the HMF data; the HMF-derived model had a PR-AUC-E of 0.27 in PCAWG; *Supplementary file 1h*). The model of biallelic *PTEN*-d in uterine cancers is primarily based on depletion of non-clustered inv. 10–100 kb, whereas both the HMF and PCAWG models of monoallelic deficiency are primarily based on depletion of non-clustered tandem duplications (10–100 kb) (*Figure 6g*). In contrast, our models of *PTEN*-d in CNS cancers from HMF were based on depletion of both non-clustered deletions 1–10 kb and non-clustered translocations (*Figure 6f*).

## Additional shortlisted gene deficiency models

We predicted monoallelic *ARID1A* LOF (PR-AUC-E = 0.208; AUROC = 0.72) by depletion of SBS signature 8 mutations in metastatic prostate cancer (*Figure 6—figure supplement 1a*). Loss of *ARID1A* impairs the pausing of RNA polymerase II during transcription, leading to dysregulated gene expression (*Trizzino et al., 2018*), whereas SBS signature 8 mutations have been associated with inactivity of *BRCA1/2* (*Davies et al., 2017*). When contrasting *ARID1A* LOF mutated patients specifically with *BRCA1/2* WT patients within the prostate cancer cohort, a significant depletion of SBS signature 8 mutations remained (Wilcoxon rank-sum test p < 0.0013), which expectedly increased when comparing with *BRCA1/2*-deficient patients (p < $7.7 \times 10^{-5}$; *Figure 6—figure supplement 2d*).

We could predict the biallelic deficiency of *BAP1* in metastatic skin cancers (PR-AUC-E = 0.26; AUROC = 0.80) primarily by a decreased number of SBS signature 7 mutations, UV induced (*Pfeifer et al., 2005*; *Howard and Tessman, 1964*), and an elevated number of SBS signature 30 mutations, related to inefficient base excision repair (*Drost et al., 2017*; *Supplementary file 1d and h*; *Figure 6—figure supplement 1b*). As the two signatures have high compositional similarity (cosine similarity = 0.72), their co-occurrence and opposite predictive effects may be caused by technical difficulties with their combined inference. *BAP1* germline variants are commonly associated with predisposition for the development of multiple cancer types including melanoma (*Pilarski et al., 2020*).

In metastatic neuro-endocrine cancers, biallelic loss of the *MEN1* gene was predicted by fewer mutations attributable to SBS signature 16, of unknown aetiology, and 9, which is related to hyperactivity of POLH (PR-AUC-E = 0.22; AUROC = 0.82) (*Figure 6—figure supplement 1c*; *Supplementary*

*file 1h*). *MEN1* is a regulator of gene transcription and germline deficiencies are causatively associated with developing Multiple Endocrine Neoplasia Type 1 (MEN1), which is a rare, hereditary tumour condition (*Chandrasekharappa et al., 1997*). However, in this setting, we observed only somatic events, with all 10 patients having an LOH combined with a pathogenic variant affecting the open reading frame. (*Supplementary file 1g*). Studies of somatic *MEN1* mutations likewise found somatic hits in *MEN1* exclusively together with LOH events, and associated the somatic LOF of *MEN1* with a different disease phenotype than that of inherited MEN1 (*Heppner et al., 1997*).

We could predict *RB1* biallelic LOF among the metastatic urinary tract cancers by an increase of mutations attributable to UV-related SBS signature 7 (PR-AUC = 0.47; AUROC = 0.84) (*Figure 6—figure supplement 1d*; *Supplementary file 1h*). SBS signature 7 is known to develop from the exposure to UV light (*Supplementary file 1e*; *Howard and Tessman, 1964*). However, the signature has been reported by COSMIC for various cancer types with no sun exposure, including cancers of the breast, ovary, pancreas, oral cavity, lung, and uterus as well as sarcomas (*Tate et al., 2019*). Although the predictive performance is considerable and significant, the accumulation of mutations in SBS signature 7 does not necessarily reveal the true aetiology of the underlying mutagenesis.

## Discussion

The cancer-specific, incomplete repair of endogenous and exogenous DNA lesions leave specific genome-wide mutational patterns. Their detection provides potentially powerful information on the fidelity of individual DNA repair pathways and response to chemo- and immunotherapies. Taking advantage of a large pan-cancer data set, our analysis shows that mutational patterns are associated with DNA repair defects across a wide range of cancers and repair mechanisms. In this study, we have contributed concrete initial predictive algorithms for mutational patterns of several DDR gene deficiencies, with potential use for clinical intervention.

The clinical scope of our approach is exemplified by recent regulatory approval for PARPi administration, which was supported by statistical predictions of HRD based on mutational patterns (*FDA approval, 2019*). We have similarly predicted *CDK12* deficiency (*CDK12*-d), which has previously been associated with a tandem duplication phenotype– with power similar to that of HRD detection. This suggests a clinical benefit of clinical application of predictive algorithms to test for *CDK12*-d, specifically when considering the application of CHK1 inhibitors in prostate, ovary, and breast cancer treatment (*Paculová et al., 2017*).

We also find high predictive power for *MSH6*-d in prostate cancers by counting deletions in repetitive DNA, also known as the MSI phenotype (*Boland and Goel, 2010*). It is common practice to search for signs of MSI in colorectal cancers, endometrial cancers, and aggressive prostate cancers. This prediction is most commonly made using a panel of repetitive DNA regions, such as the Bethesda panel (*Umar et al., 2004*). Here we demonstrated high predictive power in prostate cancers based on mutation summary statistics, which may be routinely extracted from whole-genome sequencing. The correct identification of tumours with high MSI supports the administration of immune checkpoint blockade treatment, as the mutational phenotype leads to an increased expression of neo-peptides and thus an increased sensitivity towards the immune response (*Abida et al., 2019*; *Antonarakis et al., 2019*).

Our systematic approach identified a similar model of *PTEN* LOF in uterine cancer in the PCAWG data and the HMF data; either model also having predictive power in the other data set. The re-discovery in either data set grants further trust to this model, and further experimental investigation is warranted to understand the underlying aetiology of the mutation patterns, and the potential for clinical benefits.

Recent studies have suggested the use of CDK4/6 inhibition in treating *SMARCA4*-deficient tumours (*Xue et al., 2019*). We discovered that a large subset of cancers of unknown primary (>10%) have *SMARCA4* deficiency and that these cancers can be accurately predicted from their whole-genome accumulation of SBS signature 4 or SBS signature 27 mutations. This is a clinically challenging cancer type due to the lack of a primary cancer to guide treatment and any such prediction may serve in selecting treatment for these complex cancers, given additional experimental evaluation.

Of note, the inclusion of monoallelic events allowed for larger sets of tumours with DRR gene LOF, thereby increasing the power of our study when causative associations with mutational patterns exist. This is for instance the case for *TP53*, which is known to be functionally impacted by monoallelic events (*Malcikova et al., 2009*). We shortlisted 11 models of *TP53*-d, which were all monoallelic. We further found that inclusion of monoallelic LOF events matched the predictive power for *ATRX* biallelic

LOF in CNS cancers and *HERC2* biallelic LOF in skin cancers, suggesting that single genetic hits are sufficient to affect repair and ensuing mutational patterns for these genes.

Incorrectly annotated LOF events in DDR genes may affect both the ability to discover associations with mutational patterns and the performance of any associated predictive models. On the other hand, highly stringent LOF criteria may result in true LOF events being overlooked, and create small deficiency sets with insufficient power to discover true associations. For the annotation of germline LOF events, we restricted our focus to rare variants (population frequency ≤0.5%), as more common variants are likely benign. Given the high number of variants evaluated overall, false-positive germline LOF calls are expected. However, they are unlikely to associate with specific mutational patterns and thus are unlikely to contribute significant, shortlisted false-positive predictive models of DDR deficiency. Likewise, in cancers with hypermutator phenotypes, somatic LOF events may be caused by and associated with specific mutational patterns. This leads to predictive models that in effect detect instances of reverse causality, as discussed for the monoallelic models found in colorectal cancer. We investigated whether reverse causality was a likely explanation for an association captured by a predictive model by evaluating whether the annotated LOF events matched the predictive mutational features (*Supplementary file 1g*). In general, biallelic LOF criteria are less sensitive to wrongly annotated LOF events than monoallelic criteria, as double hits are rare compared to single hits. For both mono- and biallelic predictive models, the data permutations ensure that the observed association between the DDR gene LOF events and given mutational patterns are surprising.

Our conservative LOF curation, and a lack of expression or protein-level data, is expected to cause some LOF events to be missed. This may explain a high posterior probability of a particular gene LOF in some tumours, but no evidence of genetic disruption (as seen in *Figures 5e and 6h*). This highlights the scope of mutation-pattern-based predictions, in particular in tumours without canonical DDR gene LOF events. At a very least, such tumours could be considered for further scrutiny for LOF of the individual DDR gene that they are predicted to have lost.

Our systematic approach provides a proof-of-concept that will become increasingly powerful as the available data sets increase in number and size. For the current data sets, consistent validation of detected associations was challenging due to small cohorts and differences in cancer biology. To establish the basis for any future synthetic lethality uses of our predictive models, it would be desirable to establish causal relationships. This has been beyond the scope of this study but could be achieved by whole-genome sequencing of cell lines or organoids with individual DDR gene knockout. Alternatively, the loss of DDR genes can be explored in animal models as previously done for several DDR genes including *BRCA1/2* (*Evers and Jonkers, 2006*), MMR genes (*Reitmair et al., 1995*; *Prolla et al., 1998*), and *TP53* (*Donehower, 1996*).

The current study yields a catalogue of predictive models that captures both known and novel associations between DDR gene deficiencies and mutational patterns. The included DDR genes may provide targets for research and development of treatment. With further optimisation, predictive models such as these may guide the selection of therapy by adding certainty of disabled or compromised repair deficiency phenotypes found in cancers of individual patients.

# Materials and methods
## Data
The analysis was conducted on 6098 whole cancer genomes and included a set of 6065 whole cancer genomes after filtering (see below). The data came from two independent data sets: the PCAWG (tumours = 2583; ICGC study ID. EGAS00001001692) (*ICGC/TCGA Pan-Cancer Analysis of Whole Genomes Consortium, 2020*) and the HMF (tumours = 3515; Acc. Nr. DR-044) (*Priestley et al., 2019*). The data sets contain tumours from 32 cancer types and represent diverse patient groups in terms of age, gender, and disease history. PCAWG consists of tumours both with and without metastasis, whereas HMF exclusively consists of donors with tumours showing metastatic capability. To best relate the two data sets, all tumours are catalogued by the site of primary disease. For 77 of the metastatic HMF tumours, the primary site was unknown, and these are annotated as such (*Figure 1a*; *Supplementary file 1a*).

## Curating samples

In the HMF data, we selected the earliest available sample whenever multiple samples existed for the same patient. For the PCAWG data, we used the official whitelist, obtained at the ICGC resources (https://dcc.icgc.org/pcawg), and for patients with more than one sample, we always selected the earliest, whitelisted sample available. We discarded six PCAWG samples of bone/soft tissue, as these were discarded in prior publications based on the PCAWG data set (*Degasperi et al., 2020*) leaving 2568 PCAWG samples and 3497 HMF samples across 32 sites of the body. A list of all Donor IDs, Sample IDs, and primary sites of disease may be found in *Supplementary file 1a*.

## Curating variants for LOF annotation

We annotated variants across a set of 736 genes related to the DDR (*Supplementary file 1b*). The set of genes is combined from three sources, *Knijnenburg et al., 2018*; *Pearl et al., 2015*; and *Olivieri et al., 2020*.

We filtered the downloaded variant call files (VCFs) of all samples for variants between the Ensembl (GRCH37/hg19) start and end position of each DDR gene (see coordinates in *Supplementary file 1b*). We included variants classified as PASS in the VCF files and variants of the PCAWG data set supported by at least two of our four variant callers. Furthermore, we discarded all variants which occurred in more than 200 samples across the two data sets in order to avoid noise arising from single-nucleotide polymorphisms (SNPs) called as single-nucleotide variants (SNVs), SNPs with high frequency in particular populations and possibly technical artefacts. We also discarded somatic variants with variant allele fractions below 0.2 and variants where gnomAD (V2.1.1) showed a germline population frequency above 0.5% (*Karczewski et al., 2020*).

## Annotating pathogenic variants and mutations

We annotated all variants and mutations with CADD phred scores (V1.6) in order to separate likely pathogenic variants from likely benign variants (*Rentzsch et al., 2019*). All variants with CADD phred scores of 25 or higher were considered pathogenic, whereas non-synonymous mutations with CADD phred scores below 25 and above 10 were considered VUS. Variants with CADD phred scores below 10 were considered benign. A CADD phred score of 25 is a conservative threshold (*Itan et al., 2016*), and only includes the 0.3% most-likely pathogenic variants. In addition, we annotated all variants with their status in the ClinVar database (when present). Combining ClinVar and CADD phred scores ensured that we would be able to discover associations across all DDR genes, not only genes with ClinVar annotations, which are expected to be incomplete for any given gene.

## Annotating LOH and deep deletions

We also used the copy number profiles of each sample to discover genes with bi- or monoallelic losses of parts of genes. Any overlap between a gene and a copy number loss was indicated as an LOF event, under the assumption that losing any part of the protein-coding DNA is detrimental to the complete protein product. We considered events in which the minor allele copy number was below 0.2 to be an LOH, whereas we considered events with a total tumour copy number below 0.3 (major and minor allele summarised) as deep deletions. This cutoff is adapted from the work of *Nguyen et al., 2020*.

## Annotating bi- and monoallelic gene hits

We considered genes with a single pathogenic variant, either somatic or germline, to be monoallelic hit (*n* = 25,399). In cases where the monoallelic hit is accompanied by an LOH, we considered the event a biallelic loss (*n* = 6,640). Finally, genes that were completely depleted in a sample were considered biallelic lost (*n* = 66). We did not consider a single LOH event as a monoallelic loss due to the broad impact and high frequency of such events, with ~25 times higher rates of LOH than deep deletions (*Figure 1b, c*). We have summarised the causes of LOF annotation at a per-model basis (*Supplementary file 1g*), including an annotation of the number of tumours with events hotspot locations, microhomologous DNA, and repetitive DNA. We do not disclose this information at a per-sample level in order to maintain patient privacy and data safety.

## Mutational patterns of SBSs, indels, and SVs

We summarised the genome-wide set of somatic mutations by using Signature Tools Lib, developed by *Degasperi et al., 2020*. Firstly, we used Signature Tools Lib to count single-nucleotide variants by base change and context, and assigned these counts to a set of organ-specific signatures, as recommended by *Degasperi et al., 2020*. We then have converted the organ-specific signature exposures to reference signature exposures using the conversion matrix found in the supplementary material of the paper by *Degasperi et al., 2020*. For 12 signatures (N1–N12) this was not possible, and these signatures are considered to be novel. Likewise, two signatures associated with MMR deficiency, MMR1 and MMR2, are associated with several of the COSMIC MMR signatures. For these, we preserved the signature label as given by Signature Tools Lib. For cohorts with no organ-specific signatures (cancers of unknown primary, neuroendocrine tissue, thymus, urinary tract, penis, testis, small intestine, vulva, double primary, and the adrenal gland), we have assigned mutations directly to the full set of 30 reference signatures (COSMIC signatures 1–30). Note that we excluded age-related signature 1 from the modelling, as this signature confounded by acting as a proxy for the age of the patient so that the model would be learning to differentiate old from young patients rather than patients with activity of different mutational processes or repair deficiencies. We did not exclude signature 5 although it has suggested association with age, because it also has suggested associations to the hormone receptor positive subtype of breast cancers (*Perry et al., 2022*).

Secondly, Signature Tools Lib was used to count indels: Insertions were simply counted, whereas deletions were separated by the DNA context of the deletion, being microhomologous, repetitive, or none of the two. Microhomology was defined by whether the deleted sequence was similar to the region immediately 3′ of the breakpoint, as this indicates repair by microhomology-mediated endjoining. Repetitive deletions were defined by whether there is a repeat of the indel at the 3′ end of the breakpoint (*Degasperi et al., 2020*). We decided to not further compose indels into signatures, for the ease of interpretation; the type of deletion has a clear relation to the mechanism of repair, which is what we are investigating in this paper.

Finally, we used Signature Tools Lib to count SVs. SVs were separated into two groups, clustered and non-clustered based on the average distance between rearrangements (the inter-rearrangement distance). Regions with at least 10 breakpoints having at least 10 times smaller inter-rearrangement distance than the average across the genome of that cancer were considered clusters (*Nik-Zainal et al., 2016*). SVs were further divided based on the type of mutation: Deletions, tandem duplications, inversions, and translocations. Deletions, tandem duplications, and inversions were further divided by size, with intervals being 1 to 10 kb, 10 to 100 kb, 100 kb to 1 Mb, 1 to 10 Mb, and finally mutations with lengths above 10 Mb. As with indels, we decided not to decompose SV counts into rearrangement signatures, to ease interpretability and because SV signatures are less established in the field. The per-sample exposure to each summary statistic may be seen in *Supplementary file 1c* (log transformed and scaled in *Supplementary file 1d*; see method below) and the suggested aetiology (when existing) may be seen in *Supplementary file 1e*; *Alexandrov et al., 2020*; *Tate et al., 2019*.

## Preparing subsets of data for modelling

We separated the two data sets and divided the tumours into their respective cancer-type cohorts. We stratified the data for each DDR gene; designating whether the patient had a biallelic pathogenic variant, monoallelic pathogenic variant, or no pathogenic variants. We excluded tumours with inconclusive variants (CADD phred >10, <25) within the gene. For modelling of cancers with biallelic variants in a gene, we selected all DDR gene-cohort combinations where more than five tumours had biallelic variants in the same gene (*n* = 194). Likewise, we selected all cases with more than 10 tumours with monoallelic variants (*n* = 341) (*Supplementary file 1f*). This setup means that each sample may be included in developing several models but occur only once in each model as either mutated or non-mutated. Biallelic mutated tumours were also included in the models of monoallelic loss, as we consider biallelic loss a special case of monoallelic loss.

## Generating models using LASSO regression

For each model, we calculated per-sample weights to counter the imbalance in the data, so that the weight of each sample was one minus the proportion of tumours with this mutational status (either pathogenic or non-pathogenic/WT). We then used logistic regression model with LASSO

regularisation (henceforth LASSO regression), with the alpha parameter at 1, from the R-package 'glmnet' (v4.0) (*Friedman et al., 2010*) to select features with predictive power. The LASSO regression was selected to achieve a sparse set of associated features that could be readily interpreted. This is in line with the methodology used in HRDetect by *Davies et al., 2017*.

For each of $k$ cross-fold validation sets (with $k$ being the number of bi- or monoallelic hit tumours, respectively), we ran an LASSO regression with a nested fivefold cross-validation and the assigned weights (as mentioned above to counter imbalance in the data), to produce a set of lambda values. From the LASSO regression, we selected the lambda value corresponding to the binomial deviance which is one standard deviation away from the observed minimal binomial deviance of the regression, in order to avoid overfitting (overfitting may occur if taking the lambda of the minimal binomial deviance) (*Friedman et al., 2010*). In a few cases, the LASSO regression converged to larger sets of features than what is justifiable by the low number of mutated tumours, and so we limited the number of features to one feature per 10 mutated tumours, rounded up and added one (e.g. 12 mutated tumours gives basis for maximum three features). For each of the $k$ cross-folds we used the model to predict the left-out data and used these predictions for the evaluation of the model performance.

Finally, we generated a model by including the entire data set. This was done to train the best possible model in terms of selected features and coefficients, and this is the model reported in the main figures, whereas the performance measures shown in the main figures are derived from the $k$-fold cross-validation.

## Evaluating model predictive performance

We used $k$-fold cross-validation to get a measure of the predictive performance of each model. Due to the strong imbalance in the data, we use the PR-AUC enrichment from the true-positive rate (PR-AUC-E) as our statistic of performance, while also reporting traditional AUROC scores. The true-positive rate is the baseline that would be expected for a non-informed model. By subtracting the true-positive rate from the PR-AUC of each model, we can compare the performance between models despite different sample sizes.

## Model selection using Monte Carlo simulations

We selected models for further analysis (our shortlist) based on both their PR-AUC-E performance (effect size) and their significance. Across the 535 initial models, we observed a mean PR-AUC-E of 0.04 with a standard deviation of 0.099 (90%-quantile = 0.18) (*Supplementary file 1f*). The PR-AUC-E threshold was set at 0.2 and thus roughly two standard deviations above the overall mean. To evaluate significance we used Monte Carlo simulations and generated a null-distribution for each of the 535 initial models. This was done by permuting the mutation state of the underlying tumours and then running the LASSO regression on the permuted data. We ran a minimum of 10,000 permutations ($n$) for each of the 535 models, storing the PR-AUC-E for each permutation. For models where the number ($r$) of permutations leading to PR-AUC-E values as or more extreme than the original model was smaller than five, we ran an additional 10,000 permutations, up to a maximum of 30,000 permutations. A p value ($p = (r + 1)/n$) was calculated for each model (*Supplementary file 1f*). The Benjamini–Hochberg procedure was used to control the FDR and resulting adjusted p values ($q$ values) smaller than 0.05 were shortlisted (*Supplementary file 1h*). In conclusion, 48 models of 24 genes were shortlisted for further investigation.

## Evaluating model performance in the opposite data set

For each model, we evaluated the ability to predict mutations in the opposite data set, as a way to evaluate if the model is restricted to metastatic/primary tumours or not. To do this, we identified the set of mutated and WT samples in the opposite data set and used the model on each sample to acquire a posterior probability. Based on these, we calculated the PR-AUC-E of each model in the opposite data set (*Figure 3—figure supplement 1*; *Supplementary file 1h*). We permuted the mutation status of the samples 30.000 times to evaluate the significance of the PR-AUC-E of each model.

## Survival analysis

For the 48 shortlisted models, we performed a survival analysis of the underlying data (*Figure 3— figure supplement 2*; *Supplementary file 1j*). For each model, we fitted a Cox-regression model

estimating the hazard ratio based on overall survival as a function of the mutational status of the gene. For the PCAWG data, survival times and status were made available in the most recent data release at the ICGC/PCAWG webpage (https://dcc.icgc.org/api/v1/download?fn=/PCAWG/clinical_and_histology/pcawg_donor_clinical_August2016_v9.xlsx). For samples with no registered survival time, we used the feature 'donor_interval_of_last_followup'.

For the HMF data, we acquired metadata as part of the DR-282 data package. These data included biopsy date, date of death (when applicable) and treatment end date. For deceased patients, we measured the survival time from biopsy date to date of death; for the remaining patients we measured the survival time from biopsy date to treatment end date.

## Expression analysis

For the metastatic cancers of unknown primary ($n$ = 77), we also analysed the *SMARCA4* expression patterns. RNAseq FASTQ files were obtained from the HMF (DR-282) for 58 of the tumours. Reads were pseudoaligned to GRCh37 transcripts from Gencode (V37lift37) and quantified using Kallisto version 0.48.0 (*Bray et al., 2016*). For each sample, we then extracted the protein-coding *SMARCA4* transcripts (ENST00000644737.1) and calculated an overall gene-level TPM score as the sum of the transcript-level TPM scores.

## Code availability

The code needed to reproduce the analysis will be made available at https://github.com/Simon-Grund/DDR_Predict, (copy archived at swh:1:rev:c4daf1b7a9526ea411ad763c05d0c9317b45d42e; *Sørensen, 2021*) including data pre-processing, modelling of the 535 predictive models and modelling of ≥10,000 data permutation null models for each of the 535 models.

## Acknowledgements

We thank the Pan-Cancer Analysis of Whole Genomes (PCAWG), The International Cancer Genome Consortium (ICGC), and The Cancer Genome Atlas (TCGA) for access to whole cancer genomes.This publication and the underlying study have been made possible partly on the basis of the data that Hartwig Medical Foundation and the Center of Personalised Cancer Treatment (CPCT) have made available to the study. We thank Jenny Gruhn and Amy V Kaucher for valuable comments on the manuscript. We thank Dr. Andrea Degaspari for aid in using Signature Tools Lib for summarising mutation patterns.

Funding: ERH and AS were funded by the Novo Nordisk Foundation (NNF15OC0016662), Cancer Research UK (C23210/A7574), and the Danish National Research Foundation (Center grant, DNRF115). JSP, SGS, GAP, and MHC were funded by the Independent Research Fund Denmark | Medical Sciences (8021-00419B), the Danish Cancer Society (R307-A17932), Aarhus University Research Foundation (AUFF-E-2020-6-14), a PhD stipend from Aarhus University, and a stipend from Research Foundation of Central Region Denmark (A2972).

## Additional information

### Funding

| Funder | Grant reference number | Author |
| --- | --- | --- |
| Novo Nordisk Fonden | NNF15OC0016662 | Eva R Hoffmann |
| Cancer Research UK | C23210/A7574 | Eva R Hoffmann |
| Danmarks Frie Forskningsfond | 8021-00419B | Jakob Skou Pedersen |
| Kræftens Bekæmpelse | R307-A17932 | Jakob Skou Pedersen |
| Aarhus Universitets Forskningsfond | AUFF-E-2020-6-14 | Jakob Skou Pedersen |

| Funder | Grant reference number | Author |
|---|---|---|
| Sundhedsvidenskabelige Fakultet, Aarhus Universitet | PhD stipend | Simon Grund Sørensen |
| Sundhed, Region Midtjylland | A2972 | Gustav Alexander Poulsgaard |
| Danmarks Grundforskningsfond | DNRF115 | Eva R Hoffmann |

The funders had no role in study design, data collection, and interpretation, or the decision to submit the work for publication.

## Author contributions

Simon Grund Sørensen, Conceptualization, Data curation, Formal analysis, Validation, Visualization, Methodology, Writing - original draft, Writing - review and editing; Amruta Shrikhande, Investigation, Writing - original draft; Gustav Alexander Poulsgaard, Formal analysis, Investigation, Writing - review and editing; Mikkel Hovden Christensen, Supervision, Writing - original draft; Johanna Bertl, Conceptualization, Formal analysis; Britt Elmedal Laursen, Supervision, Investigation, Writing - review and editing; Eva R Hoffmann, Conceptualization, Supervision, Funding acquisition, Writing - original draft, Project administration, Writing - review and editing; Jakob Skou Pedersen, Conceptualization, Supervision, Funding acquisition, Investigation, Writing - original draft, Project administration, Writing - review and editing

## Author ORCIDs

Simon Grund Sørensen http://orcid.org/0000-0001-5494-9375
Jakob Skou Pedersen http://orcid.org/0000-0002-7236-4001

## Ethics

We analysed data generated and made available by the Pan-Cancer Analysis of Whole Genomes (PCAWG) Consortium of the International Cancer Genome Consortium (ICGC) and The Cancer Genome Atlas (TCGA) as well as the Hartwig Medical Foundation (HMF). The research conforms to the principles of the Helsinki Declaration.

## Decision letter and Author response

Decision letter https://doi.org/10.7554/eLife.81224.sa1
Author response https://doi.org/10.7554/eLife.81224.sa2

# Additional files

## Supplementary files

• Supplementary file 1. Supplementary tables. (a) All included tumours and their primary tumour locations. (b) 736 DNA damage response (DDR) genes, hg19 coordinates, and the number of pathogenic events across 6065 cancer genomes. (c) All single-base substitution (SBS) signature contributions, indels counts, and structural variant (SV) counts, per sample. (d) All SBS signature contributions, indels counts, and SV counts, per sample, log-transformed and scaled to $z$-scores. (e) Proposed aetiologies of base substitution signatures. (f) All models ($n$ = 535). (g) Pathogenic events in each of the 535 loss-of-function (LOF)-sets. (h) Shortlisted models ($n$ = 48). (i) Correlation between features in shortlisted models. (j) Survival analysis for the shortlisted models.

• MDAR checklist

## Data availability

This study is based on analyses of human germline and cancer somatic variant data. The data sets were generated and made available by the Pan-Cancer Analysis of Whole Genomes (PCAWG) consortium and from the Hartwig Medical Foundation (HMF). The majority of the data cannot be publicly accessed as it includes protected personal data, including germline variants, which cannot be made publicly available. However, accession to the underlying data sets can be achieved through applications to ICGC/TCGA and HMF as described below. The public parts of the PCAWG data set are available at https://dcc.icgc.org/releases/PCAWG, whereas controlled files may be accessed through

applications to gbGaP and DACO, which should include a project proposal, as instructed on this site https://docs.icgc.org/pcawg/data/. The ICGC study ID of the project is EGAS00001001692. The HMF data used in this project may be found by accession code DR-044 and can be obtained by submitting an application with a project proposal to the Hartwig Medical Foundation (https://www.hartwigmedicalfoundation.nl/en). Non-personal summary data have been supplied in supplementary tables. Supplementary table (a) All included tumours and their primary tumour locations. Supplementary table (b) 736 DDR genes, hg19 coordinates and the number ofpathogenic events across 6065 cancer genomes. Supplementary table (c) All SBS signature contributions, indels counts, and SV counts, per sample. Supplementary table (d) All SBS signature contributions, indels counts, and SV counts, per sample, log-transformed and scaled to z-scores. Supplementary table (e) Proposed Etiologies of base substitution signatures. Supplementary table (f) All models (n=535). Supplementary table (g) Pathogenic events in each of the 535 LOF-sets Supplementary table (h) Short-listed models (n=48). Supplementary table (i) Correlation between features in shortlisted models. Supplementary table (j) Survival analysis for the shortlisted models. The third-party software used for data analysis includes: Pathogenicity annotation using CADD annotation software, which may be accessed at https://cadd.gs.washington.edu Signature analysis using Signature Tools Lib, which has been installed from the GitHub: https://github.com/Nik-Zainal-Group/signature.tools.lib, (copy archived at swh:1:rev:af1d46750dbc2c86a60d85ef50f19f40fa33e768) that we developed locally for the analysis can be accessed at: https://github.com/SimonGrund/DDR_Predict, (copy archived at swh:1:rev:c4daf1b7a9526ea411ad763c05d0c9317b45d42e).

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
