## [Editor Report]

This is a well-motivated study looking at the association of DNA repair deficiencies with mutational patterns. This study is of interest to the cancer genomics community and highlights how the understanding of DNA repair processes can be used in the development of novel cancer therapy, and will also be of interest to researchers in the field of genomic medicine and cancer mutagenesis. It presents predictive models with potential clinical applications that can identify patients with specific gene dysfunction based on characteristic patterns of mutation. The key findings are well supported.

---

## [Decision Letter]

**Decision letter after peer review:**

Thank you for submitting your article "Pan-cancer association of DNA repair deficiencies with whole-genome mutational patterns" for consideration by *eLife*. Your article has been reviewed by 2 peer reviewers, and the evaluation has been overseen by a Reviewing Editor and W Kimryn Rathmell as the Senior Editor. The reviewers have opted to remain anonymous.

Below you will find several suggestions for text changes that will improve the clarity of the manuscript. We encourage you to consider these suggestions and ask that you provide a response indicating how you have addressed those that you agree have merit in the text.

*Reviewer #1 (Recommendations for the authors):*

Some open questions and comments follow.

The authors mention that they excluded age-related signature 1 from the modeling as a potential confounder. Given this issue, I wondered why the clocklike signature SBS5 was not excluded for the same reason.

Monte Carlo simulation to test the performance of the lasso is mentioned twice in the main text but appears not to be described in the methods.

Regarding spurious LOFs potentially driving models in some colorectal subsets, I question why hypermutated samples were not screened ahead of modeling or further QC of variant calls from HMF and PCAWG performed.

At various points in the manuscript, the authors emphasise which models are predictive across primary and metastatic samples. This is interesting. It would have been useful to see these points consolidated in a single section of the text or a figure to allow readers to consider which deficiency models are generalizable across tumours of different metastatic potentials. Do the authors have a hypothesis as to why this might be? Where models are not generalizable is this down to insufficient power or because there is an alteration in the mode of damage associated with that deficiency in mets?

Re SMARCA4-d in tumours of unknown primary with SMARCA4 WT, is it established that there is no epigenetic silencing event in these cases?

*Reviewer #2 (Recommendations for the authors):*

A well-motivated paper using various computational analyses and a pan-cancer data set to decipher DDR genes that may be used for clinical intervention after

extensive validation.

There are a few questions that need to be addressed, as outlined below:

Despite the association between BRCA genes and mutation signature 3, previous studies have suggested that lesions in PALB2 and RAD51-related genes might contribute to mutational signature 3 activity. It would be interesting to figure out why these genes were not detected by the model.

How does the damage in DDR genes differ between patients?

I would recommend that the authors perform univariate and multivariate Cox proportional hazard models to explore associations of biomarkers (24 genes) with PFS, DSS, or OS.

---

## [Author Response]

Reviewer #1 (Recommendations for the authors):Some open questions and comments follow.The authors mention that they excluded age-related signature 1 from the modeling as a potential confounder. Given this issue, I wondered why the clocklike signature SBS5 was not excluded for the same reason.

We thank the reviewer for noting the recent updates to the COSMIC annotation of the mutational signature 5, which is now listed as potentially associated with time/age. Previously, signature five was not associated clearly with time/age, and has been suggested to be associated with other DNA damage response processes including the hormone receptor positive subtype of breast cancers. We have clarified this in the manuscript, Materials and methods subsection: “Mutational patterns of single base substitutions, indels and structural variants”:

“We did not exclude signature 5 although it has suggested association with age, because it also has suggested associations to the hormone receptor positive subtype of breast cancers(Perry et al. 2022)”

Monte Carlo simulation to test the performance of the lasso is mentioned twice in the main text but appears not to be described in the methods.

This was a textual error on our part and we thank the reviewer for making us aware the structure was unclear. The Monte Carlo simulations are described in the Materials and methods subsection: “Model selection”. To make this clear, we have retitled the subsection to: “Model selection using Monte Carlo simulations”, and updated the text to make it more evident to the reader.

Regarding spurious LOFs potentially driving models in some colorectal subsets, I question why hypermutated samples were not screened ahead of modeling or further QC of variant calls from HMF and PCAWG performed.

We agree with the reviewer that hypermutated cancers present a challenge for an automated and generic modeling procedure. In particular, they may harbor large numbers of LOF events, many of which may be inconsequential for cancer evolution. (We have now included an overview of the number of LOF events per tumour (Figure 1—figure supplement 1), which helps illustrate this.) We decided to include hypermutated tumours to ensure a uniform application and evaluation of our generic procedure, and since genetic hypermutator phenotypes are well described among colorectal cancers. Given that our procedure separates samples by cancer type, hypermutated colorectal cancers samples would not affect models trained for other cancer types.

Inspired by the comment from the reviewer, we evaluated the enrichment of LOF mutations among the subset of hypermutated tumors and now include a paragraph on this in the subsection: “Colorectal cancer models derived from hypermutated mismatch repair deficient tumours”:

“The high number of models of monoallelic deficiencies may arise from spurious LOF events in DDR genes in a subset of colorectal cancers that are hypermutated. In line with this, the hypermutated samples (n=18; >100,000 mutations) harbour 22% (5.9 fold enrichment) of all the DDR LOF events across the HMF colorectal cancer samples (n=475).”

We further reran our predictive modeling pipeline with exclusion of these hypermutated colorectal tumor samples. However, no DDR genes then had LOF sets sufficiently large to fulfill our modeling criteria (n<8 in all cases). We would thus be unable to detect the *MSH3*-d model or any of the other models among the colorectal cancer samples. We did not include these results in the manuscript.

In terms of the risk of false variant calls, our setup relies on the extensive quality control performed by the PCAWG and the Hartwig Medical Foundation and thus on the tradeoff between sensitivity and specificity chosen for their variant calling. This tradeoff relies on access to BAM files and extensive experimentation, which was not the focus of our study and thus out of scope. Importantly, mutational signatures are generally robust towards false mutation calls as long as they are not systematic; this has been extensively considered for the HMF and PCAWG data specifically (PCAWG: (Alexandrov et al. 2020); HMF: (Degasperi et al. 2020)).

At various points in the manuscript, the authors emphasise which models are predictive across primary and metastatic samples. This is interesting. It would have been useful to see these points consolidated in a single section of the text or a figure to allow readers to consider which deficiency models are generalizable across tumours of different metastatic potentials. Do the authors have a hypothesis as to why this might be? Where models are not generalizable is this down to insufficient power or because there is an alteration in the mode of damage associated with that deficiency in mets?

We agree with the reviewer that it may be of general interest to include an overview of the performance of each model across metastatic and primary tumours. We have expanded our analysis to include a unified overview of the predictive power in the opposite data set. These results are now provided in Supplementary File 1h and summarized in Figure 3—figure supplement 1. We provide introduce this analysis and the models that have significant predictive power in the opposite data in a short separate Results’ subsection: “Testing models in the opposite data set” (and include a corresponding Methods’ subsection):

“Additionally, we calculated the PR-AUC-E of each model when applied to the same cohort in the opposite data set (Figure 3—figure supplement 1). Due to the difference in biology between the two sets, and low numbers of LOF mutated samples, we did not use this as a model performance criteria but have included the PR-AUC-E values and p-values from the tests (Supplementary File 1h). We identified significant predictive power, across both metastatic and primary cancers, for deficiency models of BRCA1/2, TP53, CDK12, PTEN, ARID1A, and IDH1A. Each case is described in the respective part of the results.”

Notably, the currently available data sets are not well suited to pinpoint systematic differences between metastatic and primary tumours due to varying cohort sizes of different cancer types within and between the two data sets. Many models cannot be evaluated in other cancer types or the opposite data set due to lack of samples with mutations in a given DDR gene. Thus, we cannot exclude that more models may have predictive power between both metastatic and primary cancers than what we can currently identify.

Re SMARCA4-d in tumours of unknown primary with SMARCA4 WT, is it established that there is no epigenetic silencing event in these cases?

We do not have access to epigenetic data from the tumours of unknown primary, but were able to access expression data for a subset of them (58 of 77). Expression analysis did not indicate that epigenetic silencing was taking place, which we now show in Figure 6k and state in the subsection: “Predictive model for SMARCA4-d in cancer of unknown primary”:

“We evaluated the expression of SMARCA4 among the cancers of unknown primary and identified a significantly lower expression of SMARCA4 in tumours with biallelic LOF mutations compared to wild type (p=0.54x10^-2^; Figure 6k). For the 12 tumours with high signature 27 exposure but no biallelic LOF mutations, we did not observe a similar decrease in expression (Figure 6k), suggesting that SMARCA4 was not epigenetically silenced or otherwise transcriptionally inactivated in these cases.”

Reviewer #2 (Recommendations for the authors):A well-motivated paper using various computational analyses and a pan-cancer data set to decipher DDR genes that may be used for clinical intervention afterextensive validation.There are a few questions that need to be addressed, as outlined below:Despite the association between BRCA genes and mutation signature 3, previous studies have suggested that lesions in PALB2 and RAD51-related genes might contribute to mutational signature 3 activity. It would be interesting to figure out why these genes were not detected by the model.

We agree with the reviewer that it is reasonable to expect associations between *PALB2* and *RAD51* and signature 3. However, we did not have sufficient data to evaluate this as we only identified a single patient with a LOF mutation in *RAD51* and a single in *PALB2* across all breast cancer samples of both data sets.

How does the damage in DDR genes differ between patients?

We have included Supplementary File 1g, which summarizes the types of DDR gene damages across the samples included for the training of each predictive model. Additionally, we now include Figure 1—figure supplement 1, which shows the rate of LOF events in DDR genes at a per-tumour level. To preserve patient privacy and the data agreements, we cannot present DDR gene damages at an individual sample level. We now detail this in the Materials and methods section, subsection: “Annotating biallelic and monoallelic gene hits”:

“We have summarised the causes of LOF annotation at a per-model basis (Supplementary File 1g), including an annotation of the number of tumours with events hotspot locations, microhomologous DNA, and repetitive DNA. We do not disclose this information at a per-sample level in order to maintain patient privacy and data safety.“

I would recommend that the authors perform univariate and multivariate Cox proportional hazard models to explore associations of biomarkers (24 genes) with PFS, DSS, or OS.

We agree that it is of general interest to understand the association between DDR deficiencies and survival, though we have low numbers of LOF mutated samples in most cases as well as missing survival data in some cases, and thus limited statistical power for detecting significant hazard associations.

We have now included Kaplan-Meier plots and the results of fitting univariate Cox-regression models to overall survival statistics (the only type we have available). The results are reported in the Results’ subsection “Survival analysis” with a full summary given in Figure 3—figure supplement 1 and individual Kaplan-Meier plots shown in Figure 3—figure supplement 2:

“For each of the shortlisted models, we evaluated the difference in overall survival between samples carrying LOF mutations and those that did not. We observed nominally significant differences (p<0.05; univariate Cox regression analysis) in survival for BRCA2 and TP53 in multiple cancer types as well as for UVRAG in colorectal cancer (Figure 3—figure supplement 1,2 Supplementary file 1j). The association of TP53 monoallelic LOF with decreased survival is in line with expectations^29^. Interestingly, several models of BRCA1/2 LOF mutations associated with improved survival, including BRCA1 LOF mutations in metastatic ovary cancers (HR<0.42; p<0.093) and BRCA2 LOF mutations in non-metastatic ovary cancers (HR<0.24;p<0.017). In contrast, BRCA2 LOF mutations in primary breast cancers were associated with decreased survival (HR>9.30; p<0.004). This may potentially reflect differences in both molecular diagnostic practices and treatment regiments across these cancer types. For instance, platin-based treatment irrespective of BRCA1/2 status has been standard for groups of the ovarian and pancreatic cancer patients, while traditionally not for the breast cancer patients^30,31^. The sensitising effect of BRCA1/2 deficiency might thus explain the associated survival differences among cancer types^32^. For most models the differences in survival were insignificant, though this may be related to the generally small set of LOF mutated samples.”

We have not extended this analysis further given the general small number of LOF mutated samples and resulting issues with lack of power.